# Medical and Social Characteristics of Patients with Endometrial Hyperplasia in a Large City in Kazakhstan: A Retrospective Comparative Study

**DOI:** 10.3390/healthcare13233174

**Published:** 2025-12-04

**Authors:** Bayan Imashkyzy Imasheva, Maksut Adilkhanovich Kamaliev, Vyacheslav Notanovich Lokshin, Marina Viktorovna Kiseleva, Mariya Vladimirovna Laktionova

**Affiliations:** 1Department of Healthcare Management, Kazakhstan Medical University «KSPH», Almaty 050060, Kazakhstan; mkamaliev@mail.ru; 2Persona International Clinical Center for Reproductology, Almaty 050060, Kazakhstan; v_lokshin@persona-ivf.kz; 3Department of Oncology of Reproductive Organs, Medical Radiological Scientific Center Named After A.F.Tsyba, Obninsk 249031, Russia; kismarvic@mail.ru; 4Department of Public Health and Social Sciences, Kazakhstan Medical University «KSPH», Almaty 050060, Kazakhstan; mariya070692@gmail.com; 5Ls Clinic LLP, Almaty 050000, Kazakhstan

**Keywords:** endometrial hyperplasia, morbidity, reproductive health, risk factors, prevention, diagnosis

## Abstract

**Background/Objectives**: Endometrial hyperplasia (EH) is a pathology of the uterus, which is a pathological overgrowth of the endometrial glands associated with the risk of progression to endometrial cancer (EC). The purpose of this study was to conduct a retrospective comparative analysis of the medical and social characteristics of women with endometrial hyperplasia (EH) across two time periods (2016–2017 and 2023–2024) in Almaty, the largest city in Kazakhstan. **Methods**: A retrospective comparative analysis included 376 women (188 per period) with histologically confirmed EH treated in public and private healthcare facilities. Data were extracted from electronic medical systems (Damumed, Avicenna). Group differences were evaluated using the χ^2^ test, Fisher’s exact test, and Mann–Whitney U test. Odds ratios (OR) with 95% confidence intervals (CI) were calculated; significance was set at *p* < 0.05. **Results**: The proportion of postmenopausal women increased from 22.3% to 37.8% (OR: 2.11, 95% CI: 1.34–3.32, *p* < 0.001), and self-referrals to private clinics rose from 17.6% to 37.2% (OR: 2.79, 95% CI 1.73–4.49, *p* < 0.001). Women with higher education became more prevalent (from 26.1% to 43.6%, OR: 2.19, 95% CI: 1.42–3.39, *p* < 0.001), along with an increase in endocrine and metabolic disorders such as thyroid disease (from 4.8% to 12.2%, OR: 2.77, 95% CI: 1.25–6.16) and overweight status (from 51.6% to 65.4%, OR: 1.78, 95% CI: 1.17–2.69, *p* = 0.020). Asymptomatic cases were more frequently detected (from 18.6% to 28.2%, OR: 1.72, CI: 1.06–2.79, *p* = 0.028), and diagnostic approaches shifted from blind curettage (78.2% vs. 47.3%, OR: 0.25, CI: 0.16–0.39, *p* < 0.001) toward hysteroscopy with biopsy (from 21.3% to 53.7%, OR: 4.30, CI: 2.73–6.75, *p* < 0.001). **Conclusions**: Over seven years, the clinical and socio-demographic composition of women with EH in Almaty has changed toward older, more educated, and metabolically burdened populations, with broader access to minimally invasive diagnostic methods. The findings describe observable structural changes and risk group patterns, emphasizing the importance of prospective, registry-based, and molecularly oriented studies to refine clinical strategies for prevention and early detection.

## 1. Introduction

Endometrial hyperplasia (EH) is one of the most common pathological conditions of the endometrium and is an excessive proliferative change in the uterine mucosa caused by an imbalance of sex hormones, in particular a dominant influence of estrogen with a lack of progesterone [1]. According to epidemiological studies, about 200,000 new cases of EH are registered annually in developed countries [2]. However, obtaining reliable true estimates of morbidity remains difficult due to the heterogeneity of diagnostic criteria, differences in approaches to hormone therapy, and the possibility of combination with concomitant endometrial cancer (EC) [3]. The highest rate of EH detection is observed in women aged 50–54 years, mainly in the presence of obesity with a body mass index over 30 kg/m^2^ [4]. The leading clinical manifestation of EH is abnormal uterine bleeding, especially in the perimenopausal and postmenopausal periods, which significantly reduces the quality of life of patients [5].

In recent decades, the incidence of endometrial cancer (EC) has shown a steady upward trend, especially in high-income countries [6]. In this context, investigating the factors that precede the development of endometrial carcinoma becomes particularly relevant, including the structural and morphological alterations such as endometrial hyperplasia. According to the World Health Organization (WHO, 2014) classification developed by the International Society of Gynecological Pathologists, EH is divided into non-atypical (NAEH) and atypical (AEH) forms, which has key clinical significance [4,7]. Distinguishing between these forms determines clinical management and treatment strategy, as the presence of cytologic atypia substantially increases the risk of malignant transformation. International studies have demonstrated that EH should be regarded as a precursor of EC, which, if left untreated, may either progress to or coexist with malignancy [8,9,10]. The probability of progression from non-atypical EH to EC is estimated at less than 5% over 20 years, whereas AEH carries a risk of up to 28% [9], with concurrent carcinoma found in as many as 60% of AEH cases [10]. In a comprehensive systematic review and meta-analysis of 36 studies, Doherty et al. (2020) reported that AEH coexisted with carcinoma in 32.6% of cases (95% CI: 24.1–42.4%) and had an annual progression rate of 8.2% (95% CI: 3.9–17.3%), whereas non-atypical EH showed a much lower transformation risk—about 2.6% per year (95% CI: 0.6–10.6%) [11]. Similarly, Wang et al. (2024) summarized the cumulative risk of progression as 1–3% for NAEH and up to 29% for AEH [12].

Thus, a clear differentiation between AEH and NAEH has important clinical implications for endometrial cancer prevention: AEH requires a more active diagnostic and therapeutic approach, whereas NAEH can often be managed conservatively under careful follow-up. These distinctions reinforce the rationale for studying EH as a critical component of endometrial cancer prevention strategies.

In recent years, advances in molecular pathology have substantially refined the diagnostic approach to endometrial diseases, integrating morphological and imaging methods with molecular biomarkers to achieve more precise risk stratification. In gynecologic oncology, growing attention has been directed toward biomarkers associated with human epidermal growth factor receptor 2 (HER2) pathways, which are increasingly regarded as promising tools for refined risk assessment and personalized patient management [13]. In parallel with molecular pathology, metabolomics-based approaches have emerged as promising tools for refining endometrial cancer diagnostics. Recent tissue-based untargeted metabolomic profiling using LC-HRMS has shown that specific metabolic signatures can reliably distinguish EC, EH, and normal endometrium. Moreover, already at the stage of hyperplasia, characteristic alterations in lipid metabolism and redox-related processes are observed, indicating their potential role as early metabolic biomarkers of malignant transformation [14]. Complementary review studies indicate that metabolomics provides direct insight into the molecular landscape of EC and represents a promising approach for improving early diagnosis and risk stratification. Metabolite biomarkers from serum and tissue samples may enhance current strategies for diagnosis, prognosis, and recurrence monitoring within an omics-driven precision medicine framework [15].

In a large cohort study by Aro et al. (2025) including 1239 tumor samples, HER2 amplification was shown to occur predominantly within the p53-abnormal molecular subgroup, whereas low HER2 expression was distributed across various high-risk histological types [16]. Similarly, Kim et al. (2024) demonstrated the prognostic significance of HER2 expression and amplification and highlighted the therapeutic potential of HER2-targeted therapy [17]. Although EH itself is not a HER2-driven condition, these findings strengthen the relevance of molecular biomarkers and underscore the need to incorporate them into future research aimed at stratifying the risk of EH progression. In addition, recent real-world practice data have further reinforced the role of HER2 as an actionable biomarker in aggressive endometrial carcinoma. In a large implementation study of HER2 testing involving 192 tumor samples from 180 patients with endometrial carcinoma, HER2 positivity was detected in 28% of all cases and in 30% of tumors with aberrant p53, confirming that a substantial proportion of patients may be eligible for targeted anti-HER2 therapy [18]. Contemporary review data indicate that HER2-positive serous EC represent a high-risk subtype with an unfavorable prognosis and significant therapeutic implications. Approximately one-third of patients demonstrate HER2 overexpression, and HER2-targeted therapy—primarily based on trastuzumab—has already been introduced into clinical practice and continues to expand through the development of new agents [19].

Along with advances in the study of pathogenesis and the improvement of diagnostic approaches, the clinical and epidemiological characteristics of patients with EH in the context of demographic and social changes are of scientific and practical interest. In particular, urbanization, lifestyle, population aging and the increasing prevalence of metabolic disorders (obesity, diabetes mellitus) form an unfavorable background that enhances estrogen-dependent stimulation of the endometrium [20,21]. International data confirm that age-related peaks in the morbidity and frequency of risk factors for endometrial pathology vary depending on the socio-economic status and health system in specific regions [22]. In the Central Asian region beyond Kazakhstan, a substantial burden of uterine corpus cancer has also been observed. In Uzbekistan, corpus uteri cancer ranks among the most common female malignancies, accounting for approximately 891 new cases in 2022 (4.5% of all female cancers). In Tajikistan, there were 234 new cases (6.7%), and in Turkmenistan—164 new cases (4.4%) [23]. These data are consistent with the regional estimates from GLOBOCAN and international comparative analyses, indicating that cancers of the breast, cervix, colorectum, and uterine corpus remain the leading female cancer sites in the region [24]. Collectively, these findings emphasize the need for population-based monitoring of endometrial hyperplasia as a precursor condition and the establishment of harmonized cancer registries across Central Asian countries.

At the global level, the rising incidence of endometrial cancer further underscores the significance of this issue: according to GLOBOCAN 2022, there were more than 420,000 new cases and approximately 90,000 deaths, ranking endometrial cancer as the sixth most common malignancy among women worldwide [24]. These figures highlight the importance of detailed study of clinical and socio-medical characteristics of patients with EH across diverse populations to identify regional patterns and risk determinants.

Building upon the findings of our previous epidemiological observations, the present study was designed to conduct a comparative analysis of the clinical, social, and epidemiological characteristics of women with endometrial hyperplasia (EH) across two time periods—2016–2017 and 2023–2024. The selection of these intervals was determined by methodological and systemic considerations. The earlier period represents the stage preceding the large-scale digital transformation of Kazakhstan’s healthcare system, when electronic health information systems such as Damumed and Avicenna were only being introduced, and most clinical documentation was still maintained in paper form. The later period corresponds to the post-reform stage, characterized by the widespread use of electronic registries and improved data completeness.

Furthermore, the choice of these timeframes is supported by the results of our previous epidemiological study in Almaty [25], which revealed a sharp increase in the incidence of endometrial hyperplasia beginning in 2016. This rise coincided with the implementation of the State Program for Healthcare Development “Salamatty Kazakhstan” (2011–2015), which aimed to enhance disease prevention and screening practices through the modernization of primary healthcare services. The incidence of EH increased from 0.51 to 17.75 cases per 100,000 female population between 2012 and 2016 [25]. Thus, the years 2016–2017 represent the transitional phase before and immediately following digitalization, while 2023–2024 reflect the current stage of healthcare reforms and full integration of electronic health registries. Based on the identified epidemiological trends, the need for a more in-depth clinical analysis of patients with endometrial hyperplasia was determined. The present study aims to refine and expand earlier findings by examining the medical–social, clinical–anamnestic, and organizational–clinical characteristics of women with EH. To ensure comparability and representativeness of the results, an adapted methodology proposed by Safronov et al. [26] was applied. The scientific novelty of this research lies in a comprehensive evaluation of the dynamics of medical, social, and clinical parameters of women with EH in an urban setting, providing evidence relevant for the improvement of regional prevention programs and optimization of patient management strategies.

The purpose of this study was to conduct a retrospective comparative analysis of the medical and social characteristics of women with endometrial hyperplasia (EH) across two time periods (2016–2017 and 2023–2024) in Almaty, the largest city in Kazakhstan.

## 2. Materials and Methods

### 2.1. Study Design

This study was designed as a retrospective descriptive comparative analysis aimed at examining the medical, social, and clinical characteristics of women with endometrial hyperplasia (EH) across two independent time periods—2016–2017 and 2023–2024.

The study did not include individual longitudinal follow-up; each period represented an independent dataset.

The design and methodological framework were adapted from the approach proposed by Safronov et al. [26], who compared two temporal cohorts of women with EH. In this study, the methodology was modified for a larger sample and an expanded set of variables, allowing for a comprehensive comparison of patient characteristics within the context of a large metropolitan area.

Data were obtained from patients treated at City Clinical Hospital No. 7 (public sector) and LS Clinic (private sector) in Almaty, Kazakhstan. The same inclusion criteria, diagnostic procedures, and data registration standards were applied for both study periods to ensure comparability and minimize bias.

### 2.2. Data Sources

The material for analysis was obtained from electronic medical records extracted from the comprehensive medical information systems (CMIS) Damumed (City Clinical Hospital No. 7) and Avicenna (LS Clinic).

These systems are digital platforms integrated with the national health portals of the Ministry of Health of the Republic of Kazakhstan, ensuring the maintenance of medical documentation in a paperless format [27,28].

The use of digital records minimized the risk of missing data and improved the accuracy and reliability of the extracted information.

Data extraction and initial cleaning were performed by the research team using unified coding principles to ensure consistency between the two electronic systems. For categorical variables that were originally recorded differently in Damumed and Avicenna (e.g., education level, employment status, referral type, and selected comorbidities), a harmonized coding dictionary was created manually based on clinical meaning and documentation structure in both systems. After recoding, category distributions were reviewed to detect rare or potentially inconsistent values. When ambiguities were identified, the original electronic medical records were manually re-checked by the investigators to ensure correct category assignment.

### 2.3. Study Population

The study included all women aged 18 years and older with a histologically confirmed diagnosis of endometrial hyperplasia (ICD-10 codes N85.0—non-atypical EH, and N85.1—atypical EH).

A complete sampling approach was applied: all eligible patients with complete clinical documentation were included. The total sample comprised 376 women (188 per period).

Inclusion criteria:Female patients aged ≥ 18 years;Histologically verified EH (N85.0 or N85.1 according to ICD-10);Complete clinical and medical documentation.

Exclusion criteria:Morphologically confirmed endometrial carcinoma;EH associated with pregnancy or postpartum changes;Isolated endometrial polyps;Incomplete records or missing histological reports.

Separate analysis by histological subtype (non-atypical vs. atypical EH) was not performed due to the very low number of atypical cases (N85.1) in both samples, which precluded statistically meaningful stratification. Therefore, the comparative analysis was conducted for the study population, ensuring robust and reproducible assessment of clinical and socio-demographic trends.

### 2.4. Data Completeness and Bias Control

Only cases with fully available medical documentation were included in the analysis, ensuring completeness of key variables such as age, body mass index, reproductive history, comorbidities, ultrasound findings, and clinical presentation. All variables were standardized prior to analysis, including harmonization of data types, category labels, and value ranges across the two electronic medical record systems—Damumed (City Clinical Hospital No. 7) and Avicenna (LS Clinic). Data consistency and accuracy were verified through checks for missing, implausible, or inconsistent values. Records lacking essential variables were excluded. To reduce potential bias, identical inclusion criteria, diagnostic coding standards, and case-selection procedures were applied for both study periods (2016–2017 and 2023–2024). Potential confounders, including age, BMI, menopausal status, and socioeconomic factors, were assessed descriptively within each group to ensure comparability.

This combined approach ensured high data integrity, minimized temporal and institutional bias, and improved reproducibility of the findings.

To ensure inter-database consistency, the harmonized categorical variables were reviewed separately for each institution and study period at the stage of data preparation. In addition, a random subset of records from both Damumed and Avicenna was manually compared with the original electronic charts to verify the correctness of recoding and data transfer. This procedure did not reveal systematic discrepancies in category assignment between the two databases.

### 2.5. Histological Classification

In both study periods, histological classification of EH followed the World Health Organization criteria, distinguishing non-atypical (NAEH) and atypical (AEH) hyperplasia [4]. The Endometrial Intraepithelial Neoplasia (EIN) system was not used, as national diagnostic standards in Kazakhstan are based on WHO-2014 criteria.

All histological diagnoses were independently confirmed by at least two pathologists from the participating institutions.

### 2.6. Study Variables

The analysis included the following groups of variables:

Demographic and social: age, education, employment status, marital status, type of referral, and type of healthcare facility;

Clinical and reproductive: menstrual function, parity, contraceptive methods, gynecological diseases, and prior surgical interventions;

Somatic comorbidities: endocrine, cardiovascular, hepatobiliary, cerebrovascular, renal, and other chronic conditions;

Treatment modalities: medical therapy, curettage, hysteroscopic resection, and hysterectomy.

In the present study, the term “asymptomatic” was operationally defined as the absence of patient-reported abnormal uterine bleeding, pelvic pain, or any other gynecological complaints at the time of diagnosis, with endometrial hyperplasia being detected incidentally during routine ultrasound examination or preventive gynecological check-up.

### 2.7. Statistical Analysis

Before performing comparative analyses, the normality of quantitative variables was assessed using the Shapiro–Wilk test (for *n* < 50) and the Kolmogorov–Smirnov test (for *n* ≥ 50), as well as by evaluating skewness and kurtosis.

All quantitative variables showed a non-normal distribution; therefore, nonparametric methods (Mann–Whitney U test) were applied. Non-normally distributed data were summarized as median (Me) and interquartile range (Q1–Q3). Categorical (nominal) data were presented as absolute values (*n*) and percentages (%). No data transformation was performed, as nonparametric methods were applied for all quantitative variables.

Statistical analysis was carried out using IBM SPSS Statistics, version 27.0 (IBM Corp., Armonk, NY, USA).

Categorical variables were analyzed using the Pearson χ^2^ test or Fisher’s exact test, and quantitative variables were compared using the Mann–Whitney U test.

Results were presented as absolute counts (*n*) and percentages (%).

For variables with *p* < 0.05, odds ratios (OR) with 95% confidence intervals (CI) were additionally calculated to assess the magnitude and direction of the effect. ORs and 95% CIs were computed using the Wald method. In cases where one of the study groups contained zero events, the Haldane–Anscombe correction (+0.5 to each cell) was applied to ensure statistical stability and allow interval estimation. For such rare outcomes, the resulting ORs were interpreted cautiously due to wide confidence limits.

No multiple comparison correction was applied, as the study was descriptive and comparative in nature, focusing on clinically meaningful findings rather than exploratory hypothesis testing.

### 2.8. Ethical Approval

This retrospective study was conducted in accordance with the Declaration of Helsinki and was approved by the Local Ethics Committee of the Kazakhstan Medical University “KSPH” (Minutes No. IRB-335 dated 5 January 2023). The study used fully anonymized electronic medical records obtained from two healthcare institutions. Because no identifiable personal data were collected and no direct contact with patients occurred, individual informed consent was not required, in accordance with national regulations governing retrospective analyses of anonymized data.

## 3. Results

Prior to statistical analysis, all variables included in the Results section underwent completeness checks as part of the data preparation process. Particular attention was given to lifestyle- and history-related variables such as smoking status, body mass index (BMI), and intrauterine device (IUD) use. Records with missing values for specific variables were excluded only from the corresponding analyses, and all percentages were calculated based on the number of available observations for each parameter.

### 3.1. Socio-Demographic Characteristics of Women with EH

The study analyzed 376 case histories of patients (188 for 2016–2017 and 188 for 2023–2024) with a confirmed histological diagnosis according to ICD-10 N85—endometrial hyperplasia, who were treated at City Clinical Hospital No. 7 and the LS clinic private medical clinic in Almaty. The distribution of patients who sought medical help by age group showed statistically significant differences between the study periods (*p* = 0.026). In 2016–2017, the largest proportions of patients were in the 45–49 (31.9%, 95% CI: 24.9–39.8) and 50–54 (33.5%, 95% CI: 26.2–41.4) age groups. In 2023–2024, these proportions decreased to 23.9% (95% CI: 17.5–31.8) and 29.3% (95% CI: 22.3–37.2), respectively, while the share of older women aged 55–59 (12.2%, 95% CI: 7.6–18.6), 60–64 (10.1%, 95% CI: 5.9–16.7), and 65–69 (9.6%, 95% CI: 5.5–15.9) increased (Figure 1).

Table 1 summarizes the key socio-demographic differences between women treated for endometrial hyperplasia in 2016–2017 and 2023–2024. A statistically significant increase was observed in the proportion of self-referrals for hospital admission, which rose from 17.6% in 2016–2017 to 37.2% in 2023–2024 (OR: 2.79, 95% CI: 1.73–4.49, *p* < 0.001). Similarly, the proportion of women registered at private polyclinics increased from 17.6% to 29.8% (OR: 1.99, 95% CI: 1.22–3.25, *p* = 0.005). The share of patients with higher or incomplete higher education also increased markedly—from 26.1% to 43.6% (OR: 2.19, 95% CI: 1.42–3.39, *p* < 0.001)—while the proportion of unemployed women declined from 28.7% to 14.9% (OR: 0.43, 95% CI: 0.26–0.72, *p* = 0.001). In contrast, the number of individual entrepreneurs increased significantly from 2.7% to 11.7% (OR: 4.85, 95% CI: 1.80–13.10, *p* < 0.001), reflecting a shift toward a more economically active and privately insured patient population in the later study period.

### 3.2. General Anamnestic and Somatic Characteristics of Women with EH

Table 2 presents significant differences in the anamnestic and somatic profiles of women with EH between the two observation periods.

A statistically significant increase was observed in the proportion of patients with disorders of the thyroid gland—from 4.8% (9 of 188) in 2016–2017 to 12.2% (23 of 188) in 2023–2024 (OR: 2.77, 95% CI: 1.25–6.16, *p* = 0.010)—and in diseases of the gallbladder, biliary tract, and pancreas (from 4.8% to 10.1%; OR: 2.24, 95% CI: 0.98–5.08, *p* = 0.049).

Indicators reflecting lifestyle-related health risks also changed.

The proportion of active smokers increased from 8.5% to 16.0% (OR: 2.04; 95% CI: 1.07–3.89, *p* = 0.028), while overweight and obesity (BMI ≥ 25 kg/m^2^) became more common (51.6% vs. 65.4%, OR: 1.78, 95% CI: 1.17–2.69, *p* = 0.020).

Rare events were recorded in the later period, such as thyroid gland surgeries (from 0% to 3.2%, OR: 13.4, 95% CI: 0.75–240.1, *p* = 0.030) and positive family history of oncopathology (from 0% to 3.7%, OR: 15.6, 95% CI: 0.88–274.8, *p* = 0.015).

Given the small number of cases, these results were retained for transparency but interpreted with caution. Overall, compared with the earlier period, patients examined in 2023–2024 showed a higher prevalence of endocrine, metabolic, and lifestyle-associated conditions.

### 3.3. Reproductive and Gynecological History of Women with EH

A comparative analysis of reproductive and gynecological characteristics revealed several statistically significant differences between the two study periods. The proportion of postmenopausal women increased from 22.3% to 37.8% (OR: 2.11, 95% CI: 1.34–3.32, *p* < 0.001), indicating a higher representation of older age groups in the later period. Similarly, the frequency of women who had never been pregnant rose from 8.5% to 16.0% (OR: 2.04, 95% CI, 1.07–3.89, *p* = 0.028), and the proportion of nulliparous women increased from 12.2% to 19.7% (OR:1.76, 95% CI: 1.00–3.09, *p* = 0.049). At the same time, the proportion of patients with a history of medical abortions decreased from 53.7% to 42.0% (OR: 0.62, 95% CI: 0.42–0.94, *p* = 0.023), as did the prevalence of cervical erosion or ectropion (26.6% to 17.0%, OR: 0.57, 95% CI: 0.34–0.93, *p* = 0.025). Conversely, the prevalence of endometriosis increased from 2.7% to 9.6% (OR: 3.88, 95% CI: 1.41–10.67, *p* = 0.005), and benign mammary dysplasia from 1.6% to 9.0% (OR: 6.13, 95% CI: 1.77–21.29, *p* = 0.001). The use of intrauterine devices declined markedly—from 19.7% to 4.3% (OR: 0.18, 95% CI: 0.08–0.40, *p* < 0.001). Additionally, the proportion of blind uterine curettage procedures decreased from 66.5% to 55.9% (OR: 0.63, 95% CI: 0.41–0.97, *p* = 0.034), reflecting a gradual reduction in the use of non-visual diagnostic methods (Table 3).

Table 4 illustrates a comparison of clinical and diagnostic parameters between the two study periods, demonstrating several significant changes. The median endometrial thickness measured by transvaginal ultrasound decreased from 12.0 mm (IQR 9.7–15.0) in 2016–2017 to 10.0 mm (IQR 8.6–13.8) in 2023–2024 (*p* = 0.005), reflecting an overall trend toward earlier or preventive detection. The proportion of patients with asymptomatic presentation increased from 18.6% to 28.2% (OR: 1.72, 95% CI: 1.06–2.79, *p* = 0.028). At the same time, the frequency of blind uterine curettage significantly decreased (78.2% to 47.3%, OR: 0.25, 95% CI: 0.16–0.39, *p* < 0.001), whereas hysteroscopy with targeted biopsy became more frequent (21.3% to 53.7%, OR: 4.30, 95% CI: 2.73–6.75, *p* < 0.001). Together, these data suggest broader implementation of visually guided diagnostic procedures in recent years.

## 4. Discussion

The results of this retrospective study indicate that over the past seven years, the profile of women with EH—including medical, social, reproductive, and clinical characteristics—has undergone significant changes. These differences should be interpreted as reflections of demographic and healthcare system transformations rather than as direct causal relationships.

### 4.1. Demographic and Socio-Medical Characteristics

The age distribution of patients demonstrates a gradual shift toward older age groups: although women aged 50–54 years remained predominant in both cohorts, the proportion of those aged 55 years and older increased in 2023–2024 compared with 2016–2017. This shift aligns with the demographic aging of Kazakhstan’s population. According to the 2019 national census, the proportion of older individuals has risen over the past decade, accompanied by a decline in the working-age population [29]. In metropolitan areas such as Almaty, these processes are amplified by internal migration, concentration of healthcare facilities, and broader access to specialized diagnostics. Changes in patient referral and hospitalization patterns reflect ongoing transformations in the gynecological care system. The increase in self-referrals and the growing share of patients managed in private medical centers correspond with international evidence describing the expansion of the private healthcare sector, improved access to diagnostic services, and greater health awareness among the population [30]. These shifts are also consistent with the national program “Digital Health 2025”, aimed at integrating medical information systems and optimizing patient pathways within the framework of the Digital Kazakhstan initiative [31].

The sociodemographic profile of patients has also evolved: the proportion of women with higher education and individual entrepreneurs has increased. These changes likely reflect higher health literacy and greater readiness for self-initiated medical consultations, including preventive examinations. Similar associations between educational level and active healthcare utilization have been confirmed in international studies [32,33].

### 4.2. Medical and Metabolic Characteristics

Obesity and metabolic–endocrine disorders remain the principal risk factors for EH, primarily through mechanisms of hyperinsulinemia, hyperestrogenism, and progesterone resistance [34,35]. Comparable findings have been reported in cohort studies showing that women with BMI ≥ 30 kg/m^2^ had a fourfold higher incidence of EH and endometriosis [36,37]. This study also observed an increased frequency of hepatobiliary, pancreatic, and thyroid diseases. These comorbidities may be associated with urban lifestyle factors such as physical inactivity, high-calorie diets, and altered gallbladder motility [38,39,40]. In addition, Almaty is located in an iodine-deficient region [41], which contributes to the high prevalence of hypothyroidism and nodular goiter—conditions associated with ovulatory dysfunction and relative hyperestrogenism [42,43,44]. Collectively, these factors form an unfavorable metabolic-hormonal background that may predispose to endometrial hyperplasia.

### 4.3. Reproductive and Gynecological Characteristics

Changes in reproductive history reflect both demographic aging and evolving reproductive behavior in urban populations. The increase in postmenopausal and nulliparous women parallels the shift in disease prevalence toward older age groups. Nulliparity and low parity are well-established risk factors for EH, associated with prolonged estrogen exposure in the absence of sufficient progesterone regulation [45]. Rodrigues et al. reported similar findings [36]. The reduction in IUD use (from 19.7% to 4.3%) likely reflects the predominance of postmenopausal women in the recent cohort. However, decreased use of levonorgestrel-releasing IUDs represents an unfavorable clinical tendency, given their proven preventive and therapeutic potential in EH [46]. The decline in cervical erosion and ectropion rates is likely related to the implementation of the State Health Development Program “Densaulyk” (2016–2019), which focused on improving prevention and early detection of precancerous conditions [47]. In contrast, the increased detection of endometriosis and benign breast dysplasia probably reflects enhanced diagnostic capacity in metropolitan settings. Both conditions are characterized by a hyperestrogenic and inflammatory milieu that may contribute to the risk of EH [48,49]. The decline in blind uterine curettage and the corresponding rise in hysteroscopy with biopsy indicate the gradual adoption of visually guided diagnostic methods in line with FIGO and ACOG recommendations [50,51].

### 4.4. Diagnostic Aspects and Technological Access

The increased proportion of asymptomatic cases and decreased endometrial thickness at detection are likely associated with broader availability of transvaginal ultrasound and hysteroscopy, which have become the standard tools for evaluating intrauterine pathology [52]. The integration of these minimally invasive methods into both public and private healthcare facilities has improved the early identification of EH and the precision of targeted biopsies.

### 4.5. Current Research and Future Perspectives

Recent progress in gynecologic oncology increasingly emphasizes the use of molecular biomarkers to improve risk stratification and early detection of endometrial carcinoma. Although endometrial hyperplasia itself is not mediated by HER2-associated pathways, accumulating evidence demonstrates that HER2 amplification and overexpression play an important prognostic and therapeutic role in aggressive endometrial cancer subtypes. Large-scale molecular studies have shown that HER2 amplification is concentrated primarily within p53-abnormal carcinomas (Aro et al., 2025), while HER2-low expression is observed across several high-risk histotypes [16]. Clinical research, including the trial by Fader et al. (2020), has demonstrated that HER2-targeted therapy can improve outcomes in HER2-positive uterine serous carcinoma, reinforcing the relevance of HER2 as a clinically actionable biomarker [13]. Kim et al. (2024) further confirmed that HER2 expression and amplification independently correlate with poorer prognosis [17].

Beyond clinical trials, real-world data further support the clinical relevance of routine HER2 testing in endometrial carcinoma. Large implementation studies have demonstrated that standardized HER2 assessment in routine diagnostic practice enables identification of a substantial proportion of patients eligible for targeted anti-HER2 therapy, thus confirming the translational value of this biomarker outside controlled clinical trial settings [18]. Contemporary review data further indicate that HER2-positive serous endometrial carcinoma represents a distinct high-risk subtype with unfavorable prognosis and clinically significant therapeutic implications, with HER2-targeted treatment strategies already incorporated into current oncological practice [19].

Taken together, these findings highlight the growing role of molecular profiling in the continuum from premalignant lesions to invasive carcinoma. For research on EH, integration of molecular markers such as p53, PTEN, and HER2 into future prospective cohorts may help identify small subgroups of patients at increased risk of malignant progression. Such approaches are particularly relevant for atypical hyperplasia, where concurrent endometrial carcinoma is diagnosed in a substantial proportion of cases.

In addition to genomic and protein-based biomarkers, metabolomic profiling is increasingly recognized as a complementary tool for early detection and risk stratification in endometrial pathology. Recent tissue-based metabolomics profiling studies have demonstrated that specific metabolic signatures can distinguish not only between EC and healthy endometrium, but also between cancer and hyperplasia, supporting the concept of a metabolic continuum from precursor lesions to invasive disease [14]. Complementary review data further indicate that metabolomics biomarkers derived from serum and tissue samples may improve diagnostic accuracy, prognostic evaluation, and individualized surveillance strategies for EC [15].

In this context, establishing a biobank and expanding the citywide EH registry to include molecular annotation would enable translational studies linking clinical, histological, and molecular data. This would facilitate the development of more precise risk prediction models and support personalized surveillance strategies for women with EH.

### 4.6. Strengths and Limitations

The main strength of this study lies in the use of integrated electronic medical systems (Damumed and Avicenna), ensuring completeness, structure, and reliability of data. Comparison of two balanced cohorts provided an objective assessment of changes in patient characteristics. Limitations include data collection from only two institutions, the absence of histologic subtype stratification, and the lack of long-term outcome evaluation due to the retrospective design.

### 4.7. Future Research Directions

The findings emphasize the need for continued investigation into the epidemiological and organizational aspects of EH. Priority directions include: establishing a city-wide EH registry integrating public and private clinics; creating a biobank for molecular and metabolomic studies; conducting prospective studies assessing outcomes and quality of life; expanding research to include rural populations for urban–rural comparisons in disease detection and care accessibility. Implementation of these initiatives will provide the foundation for evidence-based strategies for the prevention and early detection of endometrial diseases in Kazakhstan.

## 5. Conclusions

This retrospective comparative study revealed that over the seven-year period (2016–2017 vs. 2023–2024), the profile of women with EH in a large urban center in Kazakhstan underwent notable demographic, clinical, and organizational changes. The increasing representation of postmenopausal women, higher educational attainment, more frequent self-referrals to private clinics, and a rise in metabolic and endocrine comorbidities reflect evolving patterns of healthcare access and population aging.

The findings illustrate epidemiological tendencies and risk group characteristics, forming a foundation for subsequent analytical and prospective research. The most vulnerable subpopulations include postmenopausal women with metabolic disorders and women without completed reproductive histories, emphasizing the need to strengthen targeted prevention, improve early diagnostic strategies, and expand access to minimally invasive techniques.

Future research should focus on prospective, multicenter, and population-based studies, as well as on the creation of regional registries and integration of biomarker and molecular data, which will support evidence-based approaches to monitoring and prevention of endometrial diseases in Kazakhstan. In the future, the integration of molecular biomarkers and metabolomic profiling into population-based clinical registries and individualized patient assessment will be essential for the successful translation of epidemiological findings into precision prevention, early diagnosis, and personalized management of EH and EC.

## Figures and Tables

**Figure 1 healthcare-13-03174-f001:**
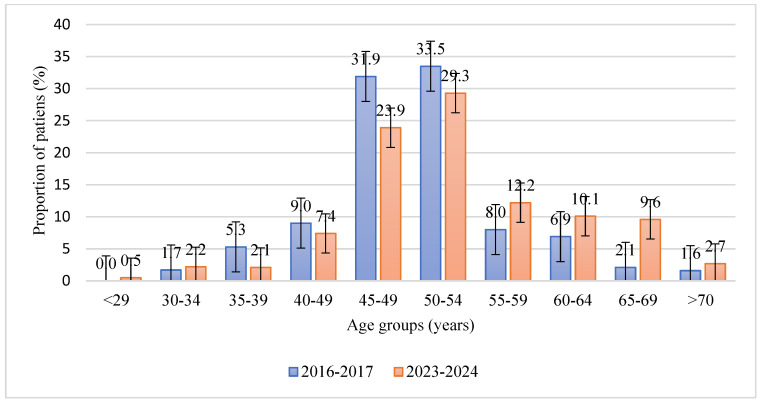
Age distribution of patients with endometrial hyperplasia in two study periods (2016–2017 and 2023–2024). Bars show the proportion of patients in each age group (%); error bars represent 95% confidence intervals. The *y*-axis is limited to non-negative values, and confidence intervals were truncated at 0% where necessary.

**Table 1 healthcare-13-03174-t001:** Socio-demographic characteristics of women with EH for the period 2016–2017 and 2023–2024.

Indicator	Study Periods		OR (95% CI)	*p*
	2016–2017 (N = 188)	2023–2024 (N = 188)		
	*n*/N (%)	*n*/N (%)		
Self-referral for hospital admission	33/188 (17.6%)	70/188 (37.2%)	2.79 (1.73–4.49)	<0.001 *
Treatment at a private policlinic	33/188 (17.6%)	56/188 (29.8%)	1.99 (1.22–3.25)	0.005 *
Higher education and incomplete higher education	49/188 (26.1%)	82/188 (43.6%)	2.19 (1.42–3.39)	<0.001 *
Individual entrepreneur	5/188 (2.7%)	22/188 (11.7%)	4.85 (1.80–13.10)	<0.001 *
Unemployed	54/188 (28.7%)	28/188 (14.9%)	0.43 (0.26–0.72)	0.001 *

Data are presented as absolute numbers (*n*) and percentage (%). Categorical variables were compared using Pearson’s χ^2^ test or Fisher’s exact test, as appropriate. Odds ratios (OR) with 95% confidence intervals (CI) were calculated using the Wald method. * The significance was set at *p*-values < 0.05.

**Table 2 healthcare-13-03174-t002:** Somatic comorbidities and lifestyle factors among women with EH for the period 2016–2017 and 2023–2024.

Indicator	Study Periods		OR (95% CI)	*p*
	2016–2017 (N = 188)	2023–2024 (N = 188)		
	*n*/N (%)	*n*/N (%)		
Diseases of the gallbladder, biliary tract and pancreas (K80–K87)	9/188 (4.8%)	19/188 (10.1%)	2.24 (0.98–5.08)	0.049 *
Disorders of the thyroid gland (E00–E07)	9/188 (4.8%)	23/188 (12.2%)	2.77 (1.25–6.16)	0.010 *
Smoking	16/188 (8.5%)	30/188 (16.0%)	2.04 (1.07–3.89)	0.028 *
BMI ≥ 25 kg/m^2^ (overweight and obesity)	97/188 (51.6%)	123/188 (65.4%)	1.78 (1.17–2.69)	0.020 *
Surgeries on the thyroid gland	0/188 (0.00%)	6/188 (3.2%)	13.4 (0.75–240.1)	0.030 *
Oncopathology	0/188 (0.00%)	7/188 (3.7%)	15.6 (0.88–274.8)	0.015 *

Data are presented as absolute numbers (*n*) and percentage (%). Categorical variables were compared using Pearson’s χ^2^ test or Fisher’s exact test, as appropriate. BMI—Body mass index. BMI category comparison: overweight + obesity vs. normal weight. Odds ratios (OR) with 95% confidence intervals (CI) were calculated using the Wald method; the Haldane–Anscombe correction (+0.5) was applied when cell counts were zero. For zero-cell outcomes, ORs were estimated with the Haldane–Anscombe correction for internal verification and are reported here for transparency, given their rarity and wide uncertainty: thyroid surgery—OR: 13.4 (95% CI: 0.75–240.1); family history of oncopathology—OR: 15.6 (95% CI: 0.88–274.8). * The significance was set at *p*-values < 0.05.

**Table 3 healthcare-13-03174-t003:** Reproductive and gynecological history of women with EH for the period 2016–2017 and 2023–2024.

Indicator	Study Periods		OR (95% CI)	*p*
	2016–2017 (N = 188)	2023–2024 (N = 188)		
	*n*/N (%)	*n*/N (%)		
Postmenopausal status (present)	42/188 (22.3%)	71/188 (37.8%)	2.11 (1.34–3.32)	<0.001 *
Absence of pregnancy (never pregnant)	16/188 (8.5%)	30/188 (16.0%)	2.04 (1.07–3.89)	0.028 *
Absence of childbirth	23/188 (12.2%)	37/188 (19.7%)	1.76 (1.00–3.09)	0.049 *
History of medical abortions	101/188 (53.7%)	79/188 (42.0%)	0.62 (0.42–0.94)	0.023 *
Erosion and ectropion of the cervix (N86)	50/188 (26.6%)	32/188 (17.0%)	0.57 (0.34–0.93)	0.025 *
Endometriosis (N80)	5/188 (2.7%)	18/188 (9.6%)	3.88 (1.41–10.67)	0.005 *
Benign mammary dysplasia (N60)	3/188 (1.6%)	17/188 (9.0%)	6.13 (1.77–21.29)	0.001 *
IUD	37/188 (19.7%)	8/188 (4.3%)	0.18 (0.08–0.40)	<0.001 *
Blind curettage performed	125/188 (66.5%)	105/188 (55.9%)	0.63 (0.41–0.97)	0.034 *

Data are presented as absolute numbers (*n*) and percentage (%). Categorical variables were compared using Pearson’s χ^2^ test or Fisher’s exact test, as appropriate. Odds ratios (OR) with 95% confidence intervals (CI) were calculated using the Wald method. IUD—Intrauterine device. * The significance was set at *p*-values < 0.05.

**Table 4 healthcare-13-03174-t004:** Clinical and diagnostic characteristics of women with EH for the period 2016–2017 and 2023–2024.

Indicator	Study Periods		OR (95% CI)	*p*
	2016–2017 (N = 188)	2023–2024 (N = 188)		
	Me (IQR)	Me (IQR)		
Endometrial thickness, M-echo (by ultrasound of pelvic organs), mm	12.0 (9.73–15.0)	10.0 (8.60–13.8)	-	0.005 *
	***n*/N (%)**	***n*/N (%)**		
Presence of asymptomatic course **	35/188 (18.6%)	53/188 (28.2%)	1.72 (1.06–2.79)	0.028 *
Curettage of the uterine cavity	147/188 (78.2%)	89/188 (47.3%)	0.25 (0.16–0.39)	<0.001 *
Hysteroscopy with targeted biopsy	40/188 (21.3%)	101/188 (53.7%)	4.30 (2.73–6.75)	<0.001 *

Data are presented as absolute numbers (*n/*N) and percentage (%), median (Me) and interquartile range (IQR). Categorical variables were compared using Pearson’s χ^2^ test or Fisher’s exact test, as appropriate. Quantitative variables were compared using the Mann–Whitney U test. Odds ratios (OR) with 95% confidence intervals (CI) were calculated using the Wald method. * The significance was set at *p*-values < 0.05. ** Under the term “asymptomatic presentation”, this study referred to cases identified during preventive ultrasound examinations or screening visits in the absence of patient complaints such as abnormal uterine bleeding or lower abdominal pain at the time of presentation.

## Data Availability

The data presented in this study can be provided by the corresponding author upon request.

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
