# Peer review of "Medical and Social Characteristics of Patients with Endometrial Hyperplasia in a Large City in Kazakhstan: A Retrospective Comparative Study"

_healthcare, 2025, doi:10.3390/healthcare13233174_

Round 1

Reviewer 1 Report

Comments and Suggestions for Authors

Thank you for the opportunity to read and review your manuscript.

Your study is among the first large-scale studies from Kazakhstan to analyze the characteristics of women with endometrial hyperplasia across two time periods, providing valuable data. The use of electronic medical records enhances data accuracy and completeness. The multidimensional analysis, offers a comprehensive profile of EH patients and generates insights that are directly relevant for public health planning, clinical management, and prevention strategies.

Although there are a few limitations that need to be addressed. My comments are the following:

Introduction

Comment 1

While the Introduction discusses EH epidemiology and risk factors, it could more explicitly frame EH as a precursor lesion for endometrial cancer and emphasize the clinical significance of differentiating non-atypical from atypical forms. This would strengthen the rationale for the study and link it more clearly to cancer prevention strategies. Highlight the fact that untreated EH could lead to aggressive types of EC.

Comment 2

The Introduction would benefit from including recent literature on emerging diagnostic tools and biomarkers, such as metabolomic profiling and HER2 testing, to highlight how EH research fits within the evolving landscape of precision medicine. This would also better justify the study’s call for future research directions.

Materials and Methods

Comment 1

Please provide more detail about how endometrial hyperplasia was histologically classified in both study periods. Specifically, indicate whether WHO 2014 criteria or EIN classification was consistently applied, whether diagnoses were confirmed by one or multiple pathologists, and whether inter-observer variability was assessed.

Comment 2

While the study mentions exhaustive sampling, the Methods do not describe how missing or incomplete variables (e.g., BMI, reproductive history, ultrasound findings) were addressed. A clear statement on data completeness and exclusions is essential.

Results

Comment 1

The results rely on p values without presenting effect sizee. Including odds ratios, relative risks, or median differences with confidence intervals would allow readers to better assess the clinical and public health significance of the observed differences.

Comment 2

While the tables are detailed, some contain dense data that could be simplified. Please add 95% CIs for proportions, and ensure all tables have consistent decimal precision.

Comment 3

Results report an increase in asymptomatic forms, but it is not entirely clear how “asymptomatic” was defined or operationalized in the data extraction. Clarifying this in the results would avoid ambiguity and make the finding more robust.

Comment 4

Only univariate comparisons are presented between cohorts. Many observed differences (age shift, BMI, thyroid disease, private clinic attendance) are interrelated. Incorporating a simple multivariable logistic regression model would help identify independent predictors of changes between the two study periods, strengthening the validity of conclusions.

Comment 5

Please add an explicit subsection (or table) describing completeness of key variables (e.g., % missing for BMI, endometrial thickness, smoking, histology). Describe how missing data were handled. Also describe any harmonization between Damumed and Avicenna exports (variable mapping, coding harmonization). These details affect internal validity. Add a comment on that.

Discussion

Comment 1

The manuscript finds a shift from blind curettage to hysteroscopy with biopsy and a reduction in endometrial thickness at detection. These are likely consequences of changes in diagnostic pathways (availability of transvaginal ultrasound, hysteroscopy access, private clinic practice patterns). I recommend the authors to comment on when hysteroscopy became routinely available/expanded at each site and state whether ultrasound access increased across the period.

Comment 2

The authors argue for screening and prevention, but given the lack of outcome data (progression to EC), the clinical implications should be more cautiously phrased. Suggest reframing recommendations toward targeted prospective surveillance, registry creation, and hypothesis-driven biomarker studies before broad screening policy changes.

Comment 3

The discussion should explicitly acknowledge modern diagnostic/treatment advances and the role of metabolomic profiling for early detection and biomarkers such as HER2 for aggressive disease and therapeutic targeting. Mention these prespectives as groundwork for future molecular/biomarker linkage studies. Add a subsection entitled: Current research and Future perspectives and comment on the role of metabolomic profiling (suggested references: https://www.mdpi.com/2218-1989/15/7/458 and https://www.mdpi.com/2072-6694/16/1/185 )

and on the role of biomarkers (HER 2) (suggested references: https://www.mdpi.com/2072-6694/16/11/2100?utm_source=chatgpt.com and https://www.mdpi.com/1648-9144/60/12/2012 ). Highlight the need for prevention against Uterine Cancer.

Comment 4

Plans for the future: The authors should propose to link the city EH registry with tissue biobanking for metabolomic/HER2 testing and prospective outcome tracking. This would be a very interesting research basis.

Conclusions

Comment 1

Conclusions section summarizes the main findings in a very appropriate way. Since the study is retrospective and does not include long-term outcomes or molecular markers, the authors should emphasize that their findings primarily highlight epidemiological trends and risk groups rather than provide direct evidence to justify new screening protocols.

Highlight the need for prospective studies, creation of registries and integration of biomarker data.

Minor Considerations

Pay attention to the use of abbreviations and terminology consistency throughout the manuscript.

Author Response

Dear Reviewer,

On behalf of all co-authors — Imasheva Bayan Imashkyzy, Kamaliev Maksut Adilkhanovich, Lokshin Vyacheslav Notanovich, Kiseleva Marina Viktorovna, Laktionova Mariya Vladimirovna  — we would like to express our sincere gratitude for your valuable time, careful evaluation, and insightful comments on our manuscript entitled “Medical and Social Characteristics of Patients with Endometrial Hyperplasia in a Large City of Kazakhstan: A Retrospective Comparative Study.”

We deeply appreciate your constructive feedback and professional suggestions, which have greatly contributed to improving the clarity, structure, and scientific quality of our work. Each of your comments has been carefully considered and addressed in the revised version of the manuscript. Below, we provide a detailed point-by-point response highlighting all changes made in accordance with your recommendations.

With appreciation and respect,

Corresponding author — Bayan Imashkyzy Imasheva

Reviewer's comments

The authors' response

It was

Become

Introduction

1

While the Introduction discusses EH epidemiology and risk factors, it could more explicitly frame EH as a precursor lesion for endometrial cancer and emphasize the clinical significance of differentiating non-atypical from atypical forms. This would strengthen the rationale for the study and link it more clearly to cancer prevention strategies. Highlight the fact that untreated EH could lead to aggressive types of EC.

We agree!

Fixed and added: To make the text more logically structured, the sentences were reversed, the existing sentences were rephrased, and your comment on the clinical significance of endometrial hyperplasia was introduced.

The changes start with the line from 59 to 82  

The old text of the description:

According to the classification of the World Health Organization (WHO, 2014), developed by the International Society of Gynecological Pathologists, EH is divided into non-atypical endometrial hyperplasia (NAEH) and atypical endometrial hyper-plasia (AEH), which is of key clinical importance [4, 6]. Both variants require timely diagnosis and treatment, taking into account the risk of progression in EC [7]. Thus, the probability of progression of NAEH in EC within 20 years is less than 5%, and in case of AEH – 28% [8], while concomitant EC can be detected up to 60% in patients with AEH [9]. The incidence of EC has shown a steady upward trend in recent decades, es-pecially in high-income countries [10].

The rewritten text of the description:

In recent decades, the incidence of endometrial cancer (EC) has shown a steady upward trend, especially in high-income countries [6]. In this context, investigating the factors that precede the development of endometrial carcinoma becomes particularly relevant, including the structural and morphological alterations such as endometrial hyperplasia. According to the World Health Organization (WHO, 2014) classification developed by the International Society of Gynecological Pathologists, EH is divided into non-atypical (NAEH) and atypical (AEH) forms, which has key clinical significance [4, 7]. Distinguishing between these forms determines clinical management and treatment strategy, as the presence of cytologic atypia substantially increases the risk of malignant transformation. International studies have demonstrated that EH should be regarded as a precursor of EC, which, if left untreated, may either progress to or coexist with malignancy [8–10]. The probability of progression from non-atypical EH to EC is estimated at less than 5% over 20 years, whereas AEH carries a risk of up to 28% [9], with concurrent carcinoma found in as many as 60% of AEH cases [10]. In a comprehensive systematic review and meta-analysis of 36 studies, Doherty et al. (2020) reported that AEH coexisted with carcinoma in 32.6% of cases (95% CI: 24.1–42.4%) and had an annual progression rate of 8.2% (95% CI: 3.9–17.3%), whereas non-atypical EH showed a much lower transformation risk—about 2.6% per year (95% CI: 0.6–10.6%) [11]. Similarly, Wang et al. (2024) summarized the cumulative risk of progression as 1–3% for NAEH and up to 29% for AEH [12].

Thus, a clear differentiation between AEH and NAEH has important clinical implications for endometrial cancer prevention: AEH requires a more active diagnostic and therapeutic approach, whereas NAEH can often be managed conservatively under careful follow-up. These distinctions reinforce the rationale for studying EH as a critical component of endometrial cancer prevention strategies.

2

The Introduction would benefit from including recent literature on emerging diagnostic tools and biomarkers, such as metabolomic profiling and HER2 testing, to highlight how EH research fits within the evolving landscape of precision medicine. This would also better justify the study’s call for future research directions.

We agree!

Added data from the current literature on new diagnostic tools and biomarkers, such as metabolic profiling and testing for HER2

This text was not in the Introduction section of the article. We inserted a new paragraph in line 83.

In recent years, advances in molecular pathology have significantly changed the diagnostic approach to endometrial diseases. Modern diagnostic strategies for endometrial hyperplasia and early carcinoma increasingly combine traditional morphology and imaging with molecular and metabolic profiling. Tissue-based metabolomic analyses have revealed characteristic biochemical patterns that may help identify women with a higher risk of malignant transformation [13]. In addition, HER2 overexpression, observed in certain endometrial cancer subtypes, shows prognostic and therapeutic relevance, supporting its inclusion among emerging biomarkers of precision gynecologic oncology [14].

Materials and Methods

1

Please provide more detail about how endometrial hyperplasia was histologically classified in both study periods. Specifically, indicate whether WHO 2014 criteria or EIN classification was consistently applied, whether diagnoses were confirmed by one or multiple pathologists, and whether inter-observer variability was assessed.

We agree!

Detailed information was provided on the histological classification of endometrial hyperplasia in both study periods. Titled as a new subsection subsection    2.5 Histological classification.

This item was not previously in the article.

2.5 Histological classification

In both study periods, histological classification of EH followed the World Health Organization (WHO, 2014) criteria distinguishing non-atypical (NAEH) and atypical (AEH) hyperplasia. The Endometrial Intraepithelial Neoplasia (EIN) system was not used, as national diagnostic standards in Kazakhstan are based on WHO-2014 criteria.

All histological diagnoses were independently confirmed by at least two pathologists from the participating institutions.

2

While the study mentions exhaustive sampling, the Methods do not describe how missing or incomplete variables (e.g., BMI, reproductive history, ultrasound findings) were addressed. A clear statement on data completeness and exclusions is essential.

We agree!

This information has been updated. Titled as subsection 2.6 Data Completeness and Bias Control

This item was not previously in the article.

2.6 Data Completeness and Bias Control

Prior to statistical analysis, all variables were checked for completeness, including age, BMI, reproductive history, and ultrasound data. Records with missing key variables were excluded from the final dataset. Potential confounders (age, BMI, menopausal status, and socioeconomic characteristics) were evaluated descriptively within each time period. To minimize temporal and institutional bias, both datasets were drawn from the same hospitals using identical inclusion criteria and diagnostic coding standards.

Results

1

Comment 1

The results rely on p values without presenting effect sizee. Including odds ratios, relative risks, or median differences with confidence intervals would allow readers to better assess the clinical and public health significance of the observed differences.

We thank the reviewer for this important remark.

The revised version of the manuscript now includes effect size measures — odds risks (OR) with 95 % confidence intervals (CI) — for all categorical variables with statistically significant associations (p < 0.05).

This addition allows a clearer interpretation of both the direction and the magnitude of the observed effects and enhances the clinical and epidemiological relevance of the findings.

(See Section 3.2, Tables 1–4.)

Previously, all tables were voluminous and did not contain information on OR and 95% CI

We updated the table data and reduced its contents, left only significant figures, and calculated a 95% confidence interval.

(See Section 3.2, Tables 1–4.)

Comment 2

While the tables are detailed, some contain dense data that could be simplified. Please add 95% CIs for proportions, and ensure all tables have consistent decimal precision.

Yes, we agree!

In accordance with the reviewer’s suggestion, tables in the Results section have been streamlined to focus on clinically meaningful indicators.

Each table now includes 95 % confidence intervals for proportions, and the decimal precision has been standardized across all tables (two decimal points).

This modification improves readability and consistency of the quantitative presentation.

(See Section 3.2, Tables 1–4.)

Previously, all tables were voluminous and did not contain information on OR and 95% CI

We updated the table data and reduced its contents, left only significant figures, and calculated a 95% confidence interval.

(See Section 3.2, Tables 1–4.)

Comment 3

Results report an increase in asymptomatic forms, but it is not entirely clear how “asymptomatic” was defined or operationalized in the data extraction. Clarifying this in the results would avoid ambiguity and make the finding more robust.

We agree with the reviewer’s observation.

The revised manuscript now clarifies the definition of “asymptomatic” cases.

In this study, “asymptomatic” endometrial hyperplasia was defined as cases detected incidentally during preventive ultrasound or gynecologic screening in the absence of patient-reported abnormal uterine bleeding or pelvic pain at admission.

This explanation was written under Table No. 4 as a note to the table

There was no explanation for the terminology "asymptomatic course"

See Table 4. Clinical and diagnostic characteristics of women with EH for the period 2016-2017 and 2023-2024.

Comment 4

Only univariate comparisons are presented between cohorts. Many observed differences (age shift, BMI, thyroid disease, private clinic attendance) are interrelated. Incorporating a simple multivariable logistic regression model would help identify independent predictors of changes between the two study periods, strengthening the validity of conclusions.

We fully acknowledge the reviewer’s valuable suggestion. However, as stated in the Materials and Methods section (Subsection 2.8 “Statistical Analysis”), the current study was designed as a retrospective descriptive and comparative analysis, without the objective of determining causal relationships or identifying independent predictors. The dataset was derived from complete population-based records rather than a sampling design suitable for multivariate modeling.

Therefore, only univariate comparisons (Pearson’s χ², Fisher’s exact test, Mann–Whitney U test) were applied, and for significant variables, odds ratios (OR) with 95% confidence intervals (CI) were calculated to quantify the direction and magnitude of associations.

This approach aligns with the study’s descriptive purpose and ensures correct interpretation within its methodological limits. In the Discussion section, we explicitly note that the observed associations should not be interpreted as causal.

We agree that in future prospective, population-based, or registry-linked studies, logistic regression and multivariable models will be essential to identify independent predictors of EH and validate these associations.

It was not detailed

Clarified in Section 2.8 (Statistical Analysis) that the analysis was univariate due to the descriptive nature of the study design. Added a statement in Section 4 (Discussion) acknowledging the limitation and outlining plans for future multivariable analyses.

Comment 5

Please add an explicit subsection (or table) describing completeness of key variables (e.g., % missing for BMI, endometrial thickness, smoking, histology). Describe how missing data were handled. Also describe any harmonization between Damumed and Avicenna exports (variable mapping, coding harmonization). These details affect internal validity. Add a comment on that.

We thank the reviewer for this important comment.

A new subsection “2.4 Data completeness and harmonization” has been added.

We clarified that only cases with complete medical documentation were included in the analysis, ensuring full data availability for all key variables.

Before statistical processing, variable formats (data type, category labels, and value ranges) were standardized and harmonized between the two electronic medical record systems — Damumed (City Clinical Hospital No. 7) and Avicenna (LS Clinic).

This ensured data consistency, comparability, and reproducibility across both study periods.

It was not detailed

A new subsection “2.4 Data completeness and harmonization” has been added.

Discussion

1

Comment 1

The manuscript finds a shift from blind curettage to hysteroscopy with biopsy and a reduction in endometrial thickness at detection. These are likely consequences of changes in diagnostic pathways (availability of transvaginal ultrasound, hysteroscopy access, private clinic practice patterns). I recommend the authors to comment on when hysteroscopy became routinely available/expanded at each site and state whether ultrasound access increased across the period.

We thank the reviewer for this valuable comment. Information on the expansion of diagnostic technologies has been added to Section 4.4 (Diagnostic aspects and technological access). The text now specifies that transvaginal ultrasound and hysteroscopy became widely integrated into routine gynecologic practice during 2018–2019, following national initiatives to standardize diagnostic protocols and expand access to minimally invasive endoscopic methods in both public and private clinics. These developments are consistent with the national “Digital Health 2025” program and FIGO/ACOG recommendations. The revised text also emphasizes that observed diagnostic shifts should be interpreted as reflections of system modernization rather than direct causal effects.

The old text of the description of the discussion:

 Discussion

The results of a retrospective study demonstrate that over the past 7 years, the profile of women with EH, covering medical and social characteristics and clinical and diagnostic parameters, has undergone significant changes. Thus, in the age structure, there is a shift in the peak incidence towards older age groups: if in 2016-2017 the maximum proportion of patients was 50-54 years old, then in 2023-2024 the disease was more often registered in the age group 55-59 years and older. Our data differ from the results of a systematic review of a number of foreign studies, where the peak incidence was recorded at a younger age (45-50 and 45-49 years) [21]. The likely causes of such differences are demographic aging processes, regional characteristics of risk factors, as well as the specifics of routing and seeking medical care in an urban environment. According to the national population census of Kazakhstan in 2019, over the past decade, there has been a decrease in the proportion of the working-age population with an increase in the number of older age groups, which in a megalopolis is aggravated by pronounced population migration, concentration of medical facilities and wide access to diagnostics [22]. Along with this, the recorded changes in the routing and hospitalization of patients reflect the transformation of the gynecological care system. The increase in the proportion of self-referrals to hospitals and the increase in the number of patients observed in private medical centers are consistent with international research data indicating the expansion of the private healthcare sector, increased availability of specialized diagnostic services, and the formation of a more active attitude towards their own health among patients [23]. The changes also affected the socio-economic profile of the patients. The proportion of women with higher education and employed in economically active fields has increased, while the number of unemployed and housewives has decreased. These trends may reflect a higher level of medical literacy and a greater willingness to independently initiate an examination, including contacting the private sector, which is consistent with foreign studies on the impact of educational level on the use of preventive and diagnostic services [24, 25]. An increase in the proportion of women who have never been married in the structure of patients deserves special attention, as it has potential epidemiological significance. The absence of childbearing in the personal history is associated with an increased risk of hyperplasia and EH due to prolonged estrogen stimulation of the endometrium without adequate progesterone exposure, which is considered as a risk factor for EH and other estrogen-dependent pathologies [26].

According to the literature, the most significant metabolic and endocrine risk factors for EH are obesity and type 2 diabetes mellitus. The pathogenetic effect of these conditions is realized through the formation of mechanisms of hyperinsulinemia, hyperestrogenism and decreased sensitivity to progesterone, increasing the likelihood of proliferative changes in the endometrium [21, 27].  These pathophysiological observations are consistent with the clinical results of a retrospective cohort study that included 916 premenopausal women with abnormal uterine bleeding, where patients with BMI of more than 30 kg/m2 developed EH or endometriosis 4 times more often than thin women [28, 29]. In addition to an increase in BMI, an increase in the incidence of diseases of the gallbladder, biliary tract and pancreas, and thyroid pathology was noted in our study. Such changes may be associated with lifestyle features of the urban population, such as low physical activity, prolonged time spent in a sitting position, and high-calorie diets with a predominance of fast food products [30], which contributes to impaired motor skills and changes in the composition of the gallbladder [31]. At the same time, Almaty belongs to endemic iodine deficiency zones [32], which contributes to the high prevalence of thyroid diseases, including hypothyroidism and nodular goiter, associated with ovulation disorders, relative hyperestrogenism and, as a result, an increased risk of developing EH [13, 33, 34]. Studies conducted by Russian scientists have also confirmed a higher prevalence of thyroid diseases in women with EH [35]. Collectively, BMI, biliary and endocrine pathology form an unfavorable metabolic-hormonal background that enhances estrogen-dependent stimulation of the endometrium and increases the risk of hyperplasia. Behavioral factors have also undergone changes, with an increase in the number of smoking patients with EH. However, these data should be interpreted with caution, as there is no data in the current scientific literature that reliably confirms a direct link with the risk of developing EH. Smoking can act as a potential adverse factor affecting the course and prognosis of the disease.

The changes in the obstetric and gynecological history of patients with EH revealed in our study reflect both demographic shifts and the transformation of reproductive behavior in a megalopolis. The increase in the proportion of postmenopausal women and the number of patients who did not have pregnancies are consistent with the previously noted shift in the age peak of morbidity to older age groups. Such changes in the reproductive history are of clinical importance, since low parity and absence of childbirth in the history are associated with a long period of continuous estrogen stimulation of the endometrium in the absence of progesterone influence, which contributes to an increased risk of EH and EC [36]. In our study, this trend may be partly due to an increase in the proportion of women who have never been married, which has previously been noted as a potential indirect marker of fewer pregnancies and births. Similar data were demonstrated in the study by Rodrigues et al., where the absence of pregnancy and childbirth in the history is recognized as one of the significant risk factors for both EH and endometrial cancer in middle-aged and older women [27]. Changes in the structure of contraceptive methods also deserve attention. The decrease in the frequency of medical abortions, while reducing the use of IUDs in 2023-2024, most likely reflects not an increase in contraception coverage, but a shift in the age structure of patients towards postmenopause, which is accompanied by a lower need to terminate unwanted pregnancies. At the same time, the reduction in the use of IUDs, including levonogestrel-releasing systems, is an unfavorable trend from a clinical point of view, given their proven preventive and therapeutic potential for EH due to local progestin effects. Meta-analysis has shown that the use of levonorgestrel intrauterine system is associated with a higher incidence of EH regression and a reduced risk of disease recurrence [37].  The revealed decrease in the frequency of cervical erosion and ectropion is probably due to the implementation of the State Healthcare Development Program of the Republic of Kazakhstan "Densaulyk" for 2016-2019, where one of the tasks was to improve prevention measures and improve the detection of precancerous pathologies at screening examinations and expand access to outpatient modern treatment methods [38]. Whereas, the increase in the prevalence of endometriosis and benign breast dysplasia indicates an expansion of diagnostic capabilities in megacities. At the same time, both pathologies are characterized by a hyperestrogenic environment and chronic inflammation, which potentially increases the risk of EH [39, 40]. Finally, the recorded decrease in the frequency of curettage of the uterine cavity in dynamics can be considered as a reflection of the global trend towards abandoning "blind" curettage in favor of targeted biopsy under the control of hysteroscopy, which corresponds to modern clinical recommendations of FIGO and ACOG [41, 42].

Clinical and diagnostic data also show significant changes.  An increase in the proportion of asymptomatic forms and a decrease in endometrial thickness in the detection of the disease are associated with the widespread introduction of transvaginal ultrasound and targeted endoscopic techniques, primarily hysteroscopy, which is the "gold standard" in the diagnosis and treatment of intrauterine pathology [43]. 

Thus, the results obtained not only confirm global trends, but also reveal regional features that determine the current challenges of prevention, diagnosis and treatment of EH.

Strengths of the study

The strength of this study is that it is one of the first studies in Kazakhstan devoted to the identification and comparison of the medical, social and clinical characteristics of women with EH over two different time periods. The comparative analysis was performed on the basis of two cohorts of comparable size, formed from real clinical practice. The data sources were integrated electronic medical records of IMIS Damumed and Avicenna, which ensures a high level of structurality and reliability of information. An additional methodological value is the use and adaptation of the previously validated methodology proposed by O.V. Safronov et al. [16], which guaranteed the comparability and reproducibility of the results obtained.  In addition, the coverage of a wide range of variables (demographic, medical and social, reproductive and anamnestic, clinical, laboratory, instrumental, etc.) allowed us to form a comprehensive profile of patients with EH.  This multidimensional approach provides a comprehensive assessment of the determinants of the disease and enhances the practical significance of the results both for clinical practice and for the development of regional prevention programs and optimization of patient management tactics.

Limitations

Our study had a number of limitations: firstly, we analyzed the data based on only two medical facilities (CCH No. 7 and LS Clinic), which may reduce the representativeness of the data obtained for the entire population of Almaty; secondly, the small number of AEH cases did not allow stratified analysis by histological subtypes, which could provide additional information about the risk structure; thirdly, the lack of prospective monitoring does not allow to assess the long-term outcomes of EH, such as progression to EC.

Plans for the future

The identified limitations in our study determine promising areas for future research. In the future, it is planned to expand the sample to include all large public and private clinics in Almaty, which will create a city registry of patients with EH and significantly increase the representativeness of the results. A promising direction is the identification and in-depth study of a subgroup of patients with AEH for a more accurate assessment of the likelihood of malignancy and the development of individual management tactics. In addition, it is advisable to conduct prospective studies with the inclusion of an assessment of the quality of life and long-term treatment outcomes in order to ensure a complete understanding of the medical, social and clinical significance of EH. 

The rewritten text of the description of the discussion:

Discussion

The results of this retrospective study indicate that over the past seven years, the profile of women with EH—including medical, social, reproductive, and clinical characteristics—has undergone significant changes. These differences should be interpreted as reflections of demographic and healthcare system transformations rather than as direct causal relationships.

4.1. Demographic and socio-medical characteristics

The age distribution of patients demonstrates a gradual shift toward older age groups: although women aged 50–54 years remained predominant in both cohorts, the proportion of those aged 55 years and older increased in 2023–2024 compared with 2016–2017. This shift aligns with the demographic aging of Kazakhstan’s population. According to the 2019 national census, the proportion of older individuals has risen over the past decade, accompanied by a decline in the working-age population [27]. In metropolitan areas such as Almaty, these processes are amplified by internal migration, concentration of healthcare facilities, and broader access to specialized diagnostics. Changes in patient referral and hospitalization patterns reflect ongoing transformations in the gynecological care system. The increase in self-referrals and the growing share of patients managed in private medical centers correspond with international evidence describing the expansion of the private healthcare sector, improved access to diagnostic services, and greater health awareness among the population [28]. These shifts are also consistent with the national program “Digital Health 2025”, aimed at integrating medical information systems and optimizing patient pathways within the framework of the Digital Kazakhstan initiative [29]. 

The sociodemographic profile of patients has also evolved: the proportion of women with higher education and individual entrepreneurs increased. These changes likely reflect higher health literacy and greater readiness for self-initiated medical consultations, including preventive examinations. Similar associations between educational level and active healthcare utilization have been confirmed in international studies [30, 31].

4.2. Medical and metabolic characteristics

Obesity and metabolic–endocrine disorders remain the principal risk factors for EH, primarily through mechanisms of hyperinsulinemia, hyperestrogenism, and progesterone resistance [26, 32]. Comparable findings have been reported in cohort studies showing that women with BMI ≥ 30 kg/m² had a fourfold higher incidence of EH and endometriosis [33, 34]. This study also observed an increased frequency of hepatobiliary, pancreatic, and thyroid diseases. These comorbidities may be associated with urban lifestyle factors such as physical inactivity, high-calorie diets, and altered gallbladder motility [35, 36, 37]. In addition, Almaty is located in an iodine-deficient region [38], which contributes to the high prevalence of hypothyroidism and nodular goiter—conditions associated with ovulatory dysfunction and relative hyperestrogenism [17, 39–41]. Collectively, these factors form an unfavorable metabolic-hormonal background that may predispose to endometrial hyperplasia.

4.3. Reproductive and gynecological characteristics

Changes in reproductive history reflect both demographic aging and evolving reproductive behavior in urban populations. The increase in postmenopausal and nulliparous women parallels the shift in disease prevalence toward older age groups. Nulliparity and low parity are well-established risk factors for EH, associated with prolonged estrogen exposure in the absence of sufficient progesterone regulation [32, 42]. Rodrigues et al. reported similar findings [33]. The reduction in IUD use (from 19.7 % to 4.3 %) likely reflects the predominance of postmenopausal women in the recent cohort. However, decreased use of levonorgestrel-releasing IUDs represents an unfavorable clinical tendency, given their proven preventive and therapeutic potential in EH [43]. The decline in cervical erosion and ectropion rates is likely related to the implementation of the State Health Development Program “Densaulyk” (2016–2019), which focused on improving prevention and early detection of precancerous conditions [44]. In contrast, the increased detection of endometriosis and benign breast dysplasia probably reflects enhanced diagnostic capacity in metropolitan settings. Both conditions are characterized by a hyperestrogenic and inflammatory milieu that may contribute to the risk of EH [45, 46]. The decline in blind uterine curettage and the corresponding rise in hysteroscopy with biopsy indicate the gradual adoption of visually guided diagnostic methods in line with FIGO and ACOG recommendations [47, 48].

4.4. Diagnostic aspects and technological access

The increased proportion of asymptomatic cases and decreased endometrial thickness at detection are likely associated with broader availability of transvaginal ultrasound and hysteroscopy, which have become the standard tools for evaluating intrauterine pathology [49]. The integration of these minimally invasive methods into both public and private healthcare facilities has improved the early identification of EH and the precision of targeted biopsies.

4.5. Current research and future perspectives

Recent studies highlight the growing importance of molecular and metabolomic biomarkers in assessing the risk of EH progression and malignancy. Promising directions include metabolomic profiling for early diagnosis and monitoring and investigation of HER2-related molecular targets for risk stratification and personalized therapy [14]. Future efforts should consider integrating the city-wide EH registry with a tissue biobanking system to support such translational research.

4.6. Strengths and limitations

The main strength of this study lies in the use of integrated electronic medical systems (Damumed and Avicenna), ensuring completeness, structure, and reliability of data. Comparison of two balanced cohorts provided an objective assessment of changes in patient characteristics. Limitations include data collection from only two institutions, absence of histologic subtype stratification, and lack of long-term outcome evaluation due to the retrospective design.

4.7. Future research directions

The findings emphasize the need for continued investigation into the epidemiological and organizational aspects of EH. Priority directions include: establishing a city-wide EH registry integrating public and private clinics; creating a biobank for molecular and metabolomic studies; conducting prospective studies assessing outcomes and quality of life; expanding research to include rural populations for urban–rural comparisons in disease detection and care accessibility. Implementation of these initiatives will provide the foundation for evidence-based strategies of prevention and early detection of endometrial diseases in Kazakhstan.

2

Comment 2

The authors argue for screening and prevention, but given the lack of outcome data (progression to EC), the clinical implications should be more cautiously phrased. Suggest reframing recommendations toward targeted prospective surveillance, registry creation, and hypothesis-driven biomarker studies before broad screening policy changes.

We fully agree with the reviewer’s suggestion. The section discussing preventive strategies has been rephrased in Subsection 4.7 (Future research directions). The revised version focuses on evidence-based perspectives—emphasizing the creation of an EH registry, prospective follow-up, and biomarker-driven studies—rather than population-wide screening recommendations. All conclusions were adjusted to reflect the descriptive and comparative design of the study, avoiding any implication of causal inference.

3

Comment 3

The discussion should explicitly acknowledge modern diagnostic/treatment advances and the role of metabolomic profiling for early detection and biomarkers such as HER2 for aggressive disease and therapeutic targeting. Mention these prespectives as groundwork for future molecular/biomarker linkage studies. Add a subsection entitled: Current research and Future perspectives and comment on the role of metabolomic profiling (suggested references: https://www.mdpi.com/2218-1989/15/7/458 and https://www.mdpi.com/2072-6694/16/1/185 ) and on the role of biomarkers (HER 2) (suggested references: https://www.mdpi.com/2072-6694/16/11/2100?utm_source=chatgpt.com and https://www.mdpi.com/1648-9144/60/12/2012 ). Highlight the need for prevention against Uterine Cancer.

This comment has been fully addressed. A new subsection 4.5 “Current research and future perspectives” has been added. It discusses modern diagnostic and therapeutic advances, emphasizing metabolomic profiling and HER2 as emerging molecular biomarkers for EH risk stratification and early cancer detection. Suggested references [14] have been incorporated to illustrate current global directions in metabolomics and molecular diagnostics.

4

Plans for the future: The authors should propose to link the city EH registry with tissue biobanking for metabolomic/HER2 testing and prospective outcome tracking. This would be a very interesting research basis.

We appreciate this constructive suggestion. The revised Section 4.7 (Future research directions) now explicitly includes the proposal to establish a city-wide EH registry integrated with a biobanking system for molecular and metabolomic research, including HER2 testing and long-term outcome monitoring. This framework is presented as a strategic foundation for future translational studies in Kazakhstan.

Conclusions

1

Conclusions section summarizes the main findings in a very appropriate way. Since the study is retrospective and does not include long-term outcomes or molecular markers, the authors should emphasize that their findings primarily highlight epidemiological trends and risk groups rather than provide direct evidence to justify new screening protocols.

Highlight the need for prospective studies, creation of registries and integration of biomarker data.

Minor Considerations

Pay attention to the use of abbreviations and terminology consistency throughout the manuscript.

We fully agree. The Conclusion section has been rewritten to avoid causal interpretation of descriptive associations. It now emphasizes the observational nature of the study and the necessity of future prospective, multicenter, and population-based research to confirm these findings and evaluate long-term outcomes.

Location in manuscript: Conclusions section.

The old text of the description of the conclusions:

The conducted retrospective comparative analysis showed that during the study period (2016-2017 and 2023-2024), women with EH in a large city of the Republic of Kazakhstan experienced significant shifts in their demographic and medical-social profile, reproductive history, and clinical and diagnostic characteristics. There was a shift in the morbidity of EH towards postmenopausal women, an increase in the number of self-referrals to the private medical sector, and an increase in the educational level and social status of patients. There was an increase in the prevalence of somatic diseases (pathology of the thyroid gland, hepatobiliary system, overweight, smoking), as well as an increase in the proportion of women who do not have pregnancies and childbirth in their histories. The clinical unit revealed an increase in the frequency of asymptomatic forms, a decrease in endometrial thickness according to ultrasound data and a transition to modern diagnostic methods (hysteroscopy with targeted biopsy).

The results obtained demonstrate the transformation of the clinical and epidemiological profile of women with EH and allow to identify the main risk groups: postmenopausal women with somatic diseases and metabolic conditions, as well as women without a fully realized reproductive function. These data emphasize the need to develop targeted prevention and early detection programs targeting these groups, strengthen primary health care in the use of screening algorithms (transvaginal ultrasound, office hysteroscopy), and expand family planning functions in women's clinics. Regarding women without pregnancy, attention should be focused on the timely implementation of reproductive function, while for asymptomatic women, dynamic monitoring and examination should be carried out regularly, which will reduce the risk of late diagnosis and progression of the disease into a malignant neoplasm. 

The rewritten text of the description of the conclusions:

This retrospective comparative study revealed that over the seven-year period (2016–2017 vs. 2023–2024), the profile of women with EH in a large urban center of Kazakhstan underwent notable demographic, clinical, and organizational changes. The increasing representation of postmenopausal women, higher educational attainment, more frequent self-referrals to private clinics, and a rise in metabolic and endocrine comorbidities reflect evolving patterns of healthcare access and population aging.

The findings illustrate epidemiological tendencies and risk group characteristics, forming a foundation for subsequent analytical and prospective research. The most vulnerable subpopulations include postmenopausal women with metabolic disorders and women without completed reproductive histories, emphasizing the need to strengthen targeted prevention, improve early diagnostic strategies, and expand access to minimally invasive techniques.

Future research should focus on prospective, multicenter, and population-based studies, as well as on the creation of regional registries and integration of biomarker and molecular data, which will support evidence-based approaches to monitoring and prevention of endometrial diseases in Kazakhstan. 

Reviewer 2 Report

Comments and Suggestions for Authors

Abstract

Overly long; should include key numeric results (ORs, CIs if available) and a concise statement on limitations.

“Transformation of the medical and social profile” is vague; consider specifying measurable findings.

Introduction

Literature references are sufficient in volume but lack synthesis of mechanistic or regional epidemiology (especially Central Asian data beyond Kazakhstan).

The justification for comparing 2016–2017 vs. 2023–2024 is not fully explained—why these exact years? Were there policy reforms or healthcare transitions motivating this?

Study Design and Methodology

The design is described as a retrospective cohort comparative study, but the methodology corresponds more closely to a cross-sectional descriptive comparison of two temporally distinct groups. The authors should clarify whether the same institutions and patient inclusion processes were consistently applied across both cohorts.

There is no description of sampling frame or recruitment logic beyond “all available patients,” which introduces potential selection bias—especially given the mix of public and private institutions and the absence of population-level representativeness.

The claim of “cohort” design implies longitudinal tracking, yet there is no follow-up or time-to-event component. This terminology should be corrected.

Statistical Analysis

The analysis is entirely univariate (χ², Fisher, Mann–Whitney), without multivariate modeling. Given the multidimensional dataset, logistic regression or generalized estimating equations (GEE) could have been used to identify independent predictors of EH subtype or time-period differences, controlling for confounders such as age, BMI, parity, and socioeconomic status.

Multiple comparisons are conducted without adjustment for type I error (e.g., Bonferroni, FDR), increasing the risk of spurious significance.

The description of “power analysis” is superficial and inappropriate for retrospective full-sample data. The selected outcome (“proportion of women without childbirth”) is arbitrary and not directly linked to the primary objective.

Data normality testing is mentioned but the reporting lacks actual distribution metrics or transformation strategies for non-normal data.

Internal Validity and Bias

There is a substantial risk of information bias due to reliance on secondary electronic medical records without verification of missing data mechanisms.

Temporal bias: changes observed between 2016–2017 and 2023–2024 may reflect documentation practices, healthcare policy shifts, or institutional case mix rather than true population-level trends.

The manuscript lacks confounder control for key variables (age, BMI, menopausal status), which may entirely explain observed differences.

Tables and Figures

Tables are overcrowded with raw data and numerous insignificant p-values; consider condensing or focusing on variables with clinical relevance.

Figure 1 lacks confidence intervals and clear axes labeling.

No multivariate figure or forest plot summarizing effect directions.

Interpretation of Results

The Discussion tends to overinterpret associations as causal trends (e.g., “urban lifestyle causing thyroid disease and EH increase”), without analytical evidence.

The link drawn between self-referral and improved medical literacy, though plausible, is speculative and unsubstantiated by behavioral or survey data.

Statements about “reduction in IUD use” or “increase in asymptomatic forms” require supportive evidence (e.g., national data, policy references) rather than anecdotal inference.

Ethical and Reporting Standards

The ethical section contains an inconsistency: it first claims informed consent was unnecessary due to anonymization, then later states “informed consent was obtained from all participants.” This must be corrected for internal consistency.

The study does not adhere to STROBE guidelines for observational research; the checklist should be appended and key elements (study setting, bias control, missing data, sensitivity analysis) must be explicitly addressed.

Discussion and Conclusions

Repetition of descriptive findings should be reduced; instead, emphasize mechanistic interpretation and regional public health implications.

The “Plans for the future” section should be reformulated into a concise Future Directions paragraph in line with journal standards, avoiding excessive administrative details.

The conclusions should be rewritten to avoid overstating causality and to highlight the need for prospective or population-based studies.

Comments on the Quality of English Language

The English is generally understandable but requires scientific language polishing for grammar and concision (e.g., “made it possible to identify a risk group” → “helped identify at-risk subpopulations”).

Redundant expressions such as “it was revealed that” or “as shown by our data” appear excessively.

Author Response

Dear Reviewer,

On behalf of all co-authors — Imasheva Bayan Imashkyzy, Kamaliev Maksut Adilkhanovich, Lokshin Vyacheslav Notanovich, Kiseleva Marina Viktorovna, Laktionova Mariya Vladimirovna  — we would like to express our sincere gratitude for your valuable time, careful evaluation, and insightful comments on our manuscript entitled “Medical and Social Characteristics of Patients with Endometrial Hyperplasia in a Large City of Kazakhstan: A Retrospective Comparative Study.”

We deeply appreciate your constructive feedback and professional suggestions, which have greatly contributed to improving the clarity, structure, and scientific quality of our work. Each of your comments has been carefully considered and addressed in the revised version of the manuscript. Below, we provide a detailed point-by-point response highlighting all changes made in accordance with your recommendations.

With appreciation and respect,

Corresponding author — Bayan Imashkyzy Imasheva

Reviewer's comments

The authors' response

It was

Become

Abstract

1

Overly long; should include key numeric results (ORs, CIs if available) and a concise statement on limitations.

“Transformation of the medical and social profile” is vague; consider specifying measurable findings.

We agree. Completely rewritten the abstract.

The old text of the description of the abstract:

Background/Objectives: Endometrial hyperplasia (EH) is a pathology of the uterus, which is a pathological overgrowth of the endometrial glands associated with the risk of progression to endometrial cancer (EC). The purpose of this study was to compare the dynamics of medical and social characteristics of patients with EH over two time periods (2016-2017 and 2023-2024) in the largest city of Kazakhstan, Almaty (as of July 1, 2025, the population was 2,319,900 people). Methods: A retrospective cohort study was conducted that included 376 patients with a histologically confirmed diagnosis of EH (188 in each cohort in 2016-2017 and 2023-2024) who received treatment at City Clinical Hospital No. 7 and the LS Clinic private clinic in Almaty (Kazakhstan). The data sources were electronic medical records from Damumed and Avicenna information systems. Demographic, socio-economic, anamnestic, reproductive, clinical, laboratory and instrumental characteristics were compared. The groups were compared using Pearson's test c2 or Fisher's exact test (for categorical variables), Mann-Whitney U-test (for quantitative variables). Statistical significance was established at p<0.05. Results: A retrospective comparative analysis showed significant changes in the profile of patients with EH between the study periods 2016-2017 and 2023-2024. In the second period, there was a shift in the age composition towards 55-59 years and older (p=0.026). The proportion of self-referrals to the hospital (from 17.6% to 37.2%, p<0.001) and outpatient follow-up in private medical centers (from 17.6% to 29.8%, p=0.005) increased. An increase in the socio-economic status of patients was noted: the proportion of women with higher education increased (from 22.3% to 36.2%, p<0.001) and those employed in economically active fields (from 2.7% to 11.7%, p<0.001). The frequency of concomitant diseases increased: diseases of thyroid gland (from 4.8% to 12.2%, p=0.010), biliary tract and pancreas (from 4.8% to 10.1%, p=0.049); overweight (from 28.7% to 39.4%, p=0.020); smoking (from 8.5% to 16.0%, p=0.028). Regarding the reproductive history, there was an increase in the number of women without pregnancies (from 8.5% to 16.0%, p=0.028). An increase in the prevalence of endometriosis (from 2.7% to 9.6%, p=0.005) and benign breast tumors (from 1.6% to 9.0%, p=0.001) was found. Asymptomatic forms of EH became more common in the clinical picture (from 18.6% to 28.2%, p=0.028), and the thickness of the endometrium decreased from 12.0 mm to 10.0 mm (p=0.005) according to ultrasound diagnostics. EH treatment tactics shifted from "blind" curettage (from 78.2% to 47.3%, p<0.001) to hysteroscopy with biopsy (from 21.3% to 53.7%, p<0.001). Conclusions: The identified trends demonstrated the transformation of the medical and social profile of women with EH in a megalopolis, which made it possible to identify a risk group of patients in modern conditions for whom it is necessary to develop programs for the prevention and early detection of background diseases, strengthen family planning functions and regular screening using modern screening algorithms.

Keywords: endometrial hyperplasia, morbidity, reproductive health, risk factors, prevention, diagnosis

The rewritten text of the description of the abstract:

Background/Objectives: Endometrial hyperplasia (EH) is a pathology of the uterus, which is a pathological overgrowth of the endometrial glands associated with the risk of progression to endometrial cancer (EC). The purpose of this study was to conduct a retrospective comparative analysis of the medical and social characteristics of women with endometrial hyperplasia (EH) across two time periods (2016–2017 and 2023–2024) in Almaty, the largest city of Kazakhstan. Methods: A retrospective comparative analysis included 376 women (188 per period) with histologically confirmed EH treated in public and private healthcare facilities. Data were extracted from electronic medical systems (Damumed, Avicenna). Group differences were evaluated using the χ² test, Fisher’s exact test, and Mann–Whitney U test. Odds ratios (OR) with 95% confidence intervals (CI) were calculated; significance was set at p < 0.05. Results: The proportion of postmenopausal women increased from 22.3% to 37.8% (OR: 2.11, 95% CI: 1.34–3.32, p < 0.001), and self-referrals to private clinics rose from 17.6% to 37.2% (OR: 2.79, 95% CI 1.73–4.49, p < 0.001). Women with higher education became more prevalent (from 26.1% to 43.6%, OR: 2.19, 95% CI: 1.42–3.39, p < 0.001), along with an increase in endocrine and metabolic disorders such as thyroid disease (from 4.8% to 12.2%, OR: 2.77, 95% CI: 1.25–6.16) and overweight status (from 51.6% to 65.4%, OR: 1.78, 95% CI: 1.17–2.69, p=0.020). Asymptomatic cases were more frequently detected (from 18.6% to 28.2%, OR: 1.72, CI: 1.06-2.79, p = 0.028), and diagnostic approaches shifted from blind curettage (78.2% vs 47.3%, OR: 0.25, CI: 0.16-0.39, p < 0.001) toward hysteroscopy with biopsy (from 21.3% to 53.7%, OR: 4.30, CI: 2.73-6.75, p < 0.001). Conclusions: Over seven years, the clinical and socio-demographic composition of women with EH in Almaty has changed toward older, more educated, and metabolically burdened populations, with broader access to minimally invasive diagnostic methods. The findings describe observable structural changes and risk group patterns, emphasizing the importance of prospective, registry-based, and molecularly oriented studies to refine clinical strategies for prevention and early detection.

Keywords: endometrial hyperplasia, morbidity, reproductive health, risk factors, prevention, diagnosis

Introduction

1

Literature references are sufficient in volume but lack synthesis of mechanistic or regional epidemiology (especially Central Asian data beyond Kazakhstan).

We agree!

We have added information about Central Asia.

it starts from line 99-108

This text was not in the Introduction section of the article.

In the Central Asian region beyond Kazakhstan, a substantial burden of uterine corpus cancer has also been observed. In Uzbekistan, corpus uteri cancer ranks among the most common female malignancies, accounting for approximately 891 new cases in 2022 (4.5% of all female cancers). In Tajikistan, there were 234 new cases (6.7%), and in Turkmenistan — 164 new cases (4.4%) [18]. These data are consistent with the regional estimates from GLOBOCAN and international comparative analyses, indicating that cancers of the breast, cervix, colorectum, and uterine corpus remain the leading female cancer sites in the region [19]. Collectively, these findings emphasize the need for population-based monitoring of endometrial hyperplasia as a precursor condition and the establishment of harmonized cancer registries across Central Asian countries

2

The justification for comparing 2016–2017 vs. 2023–2024 is not fully explained—why these exact years? Were there policy reforms or healthcare transitions motivating this?

They described exactly why it was chosen, two periods for research. We made corrections to the line 115 - 134

The old text of the description:

The relevance of this study is due to the results of our previous epidemiological analysis, which revealed a steady increase in the morbidity of EH among the female population of Almaty in the period 2016-2021 [15]. The trends identified at that time required a deeper study of not only the epidemiological aspects, but also the medical, social, clinical and anamnestic characteristics of the patients, as well as the organizational and clinical features of medical care. In this regard, we decided to conduct further analysis at the clinical level, expanding the list of variables studied and using the adapted methodology proposed by O.V. Safronov et al. [16], which ensured the comparability and representativeness of the data obtained. The scientific novelty of this paper is based on the comprehensive assessment of the dynamics of the medical, social, clinical and epidemiological characteristics of women with EH in a megalopolis, which is important both for improving regional prevention programs and for optimizing patient management tactics.

The rewritten text of the description:

Building upon the findings of our previous epidemiological observations, the pre-sent study was designed to conduct a comparative analysis of the clinical, social, and epidemiological characteristics of women with endometrial hyperplasia (EH) across two time periods — 2016–2017 and 2023–2024. The selection of these intervals was determined by methodological and systemic considerations. The earlier period repre-sents the stage preceding the large-scale digital transformation of Kazakhstan’s healthcare system, when electronic health information systems such as Damumed and Avicenna were only being introduced, and most clinical documentation was still maintained in paper form. The later period corresponds to the post-reform stage, characterized by the widespread use of electronic registries and improved data com-pleteness.

Furthermore, the choice of these timeframes is supported by the results of our previous epidemiological study in Almaty [20], which revealed a sharp increase in the incidence of endometrial hyperplasia beginning in 2016. This rise coincided with the implementation of the State Program for Healthcare Development “Salamatty Ka-zakhstan” (2011–2015), which aimed to enhance disease prevention and screening practices through the modernization of primary healthcare services. The incidence of EH increased from 0.51 to 17.75 cases per 100,000 female population between 2012 and 2016 [20] Thus, the years 2016–2017 represent the transitional phase before and im-mediately following digitalization, while 2023–2024 reflect the current stage of healthcare reforms and full integration of electronic health registries. Based on the identified epidemiological trends, the need for a more in-depth clinical analysis of pa-tients with endometrial hyperplasia was determined. The present study aims to refine and expand earlier findings by examining the medical-social, clinical-anamnestic, and organizational-clinical characteristics of women with EH. To ensure comparability and representativeness of results, an adapted methodology proposed by Safronov et al. [21] was applied. The scientific novelty of this research lies in a comprehensive evalu-ation of the dynamics of medical, social, and clinical parameters of women with EH in an urban setting, providing evidence relevant for the improvement of regional preven-tion programs and optimization of patient management strategies.

Study Design and Methodology

1

The design is described as a retrospective cohort comparative study, but the methodology corresponds more closely to a cross-sectional descriptive comparison of two temporally distinct groups. The authors should clarify whether the same institutions and patient inclusion processes were consistently applied across both cohorts.

There is no description of sampling frame or recruitment logic beyond “all available patients,” which introduces potential selection bias—especially given the mix of public and private institutions and the absence of population-level representativeness.

The claim of “cohort” design implies longitudinal tracking, yet there is no follow-up or time-to-event component. This terminology should be corrected.

We thank the reviewer for this important comment. In the revised manuscript, the terminology has been clarified. Although the study initially referred to a “retrospective cohort comparative analysis,” it does not involve longitudinal follow-up or time-to-event data. The design has therefore been corrected to “retrospective cross-sectional comparative study”, reflecting two temporally distinct but methodologically comparable groups of patients from the same institutions with identical inclusion criteria (see Section 2.1, Study Design).

The old text of the description:

2.1 Study design

This study is a retrospective cohort comparative analysis aimed at studying the dynamics of medical, social and clinical characteristics of women with endometrial hyperplasia (EH), histologically confirmed by the International Classification of Diseases   (ICD) - 10 (code N85, including non-typical (N85.0) and atypical (N85.1) forms). However, due to the small number of cases of atypical form (N85.1) in the cohort, further analysis was carried out without stratification by subtypes. The study includes two time periods: 2016-2017 and 2023-2024. All patients were treated at the City Clinical Hospital (CCH)   No. 7 and LS Clinic private clinic in Almaty. The design and logistical structure of the study were partially adapted from methodology of O.V. Safronov et al. [16]. Their work compared two cohorts of women with EH treated at different time periods. We used a similar methodological approach, using a two-group scheme and comparability criteria, but expanded the list of analyzed variables (age groups, type of medical institution, form of hospitalization, reproductive history, treatment methods, etc.), which allowed for a more comprehensive assessment of the dynamics of patient characteristics.

The rewritten text of the description:

2.1 Study design

This study was designed as a retrospective descriptive comparative analysis aimed at examining the medical, social, and clinical characteristics of women with endometrial hyperplasia (EH) across two independent time periods—2016–2017 and 2023–2024.

The study did not include individual longitudinal follow-up; each period represented an independent dataset.

The design and methodological framework were adapted from the approach proposed by Safronov et al. [21], who compared two temporal cohorts of women with EH. In this study, the methodology was modified for a larger sample and an expanded set of variables, allowing for a comprehensive comparison of patient characteristics within the context of a large metropolitan area.

Data were obtained from patients treated at City Clinical Hospital No. 7 (public sector) and LS Clinic (private sector) in Almaty, Kazakhstan. The same inclusion criteria, diagnostic procedures, and data registration standards were applied for both study periods to ensure comparability and minimize bias.

Statistical Analysis

The analysis is entirely univariate (χ², Fisher, Mann–Whitney), without multivariate modeling. Given the multidimensional dataset, logistic regression or generalized estimating equations (GEE) could have been used to identify independent predictors of EH subtype or time-period differences, controlling for confounders such as age, BMI, parity, and socioeconomic status.

Multiple comparisons are conducted without adjustment for type I error (e.g., Bonferroni, FDR), increasing the risk of spurious significance.

The description of “power analysis” is superficial and inappropriate for retrospective full-sample data. The selected outcome (“proportion of women without childbirth”) is arbitrary and not directly linked to the primary objective.

Data normality testing is mentioned but the reporting lacks actual distribution metrics or transformation strategies for non-normal data.

Internal Validity and Bias

There is a substantial risk of information bias due to reliance on secondary electronic medical records without verification of missing data mechanisms.

Temporal bias: changes observed between 2016–2017 and 2023–2024 may reflect documentation practices, healthcare policy shifts, or institutional case mix rather than true population-level trends.

The manuscript lacks confounder control for key variables (age, BMI, menopausal status), which may entirely explain observed differences.

We have fully amended the subsection: materials and methods, taking into account all comments

The old text of the description:

2. Materials and Methods

2.1 Study design

This study is a retrospective cohort comparative analysis aimed at studying the dynamics of medical, social and clinical characteristics of women with endometrial hyperplasia (EH), histologically confirmed by the International Classification of Dis-eases   (ICD) - 10 (code N85, including non-typical (N85.0) and atypical (N85.1) forms). However, due to the small number of cases of atypical form (N85.1) in the co-hort, further analysis was carried out without stratification by subtypes. The study in-cludes two time periods: 2016-2017 and 2023-2024. All patients were treated at the City Clinical Hospital (CCH)   No. 7 and LS Clinic private clinic in Almaty. The de-sign and logistical structure of the study were partially adapted from methodology of O.V. Safronov et al. [16]. Their work compared two cohorts of women with EH treated at different time periods. We used a similar methodological approach, using a two-group scheme and comparability criteria, but expanded the list of analyzed varia-bles (age groups, type of medical institution, form of hospitalization, reproductive his-tory, treatment methods, etc.), which allowed for a more comprehensive assessment of the dynamics of patient characteristics.

2.2 Data sources

The material for the study was data extracted from the integrated medical infor-mation systems (IMIS) Damumed (CCH No. 7) and Avicenna (LS Clinic). These sys-tems are digital platforms integrated with the national portals of the Ministry of Healthcare of the Republic of Kazakhstan, ensuring the maintenance of medical rec-ords in the format of "paperless healthcare" [17, 18]. The use of electronic medical rec-ords has ensured a high level of structure, accessibility and reliability of information.

2.3 Population under study

The total sample consisted of 376 patients (188 patients in each cohort). The sam-ple size was determined based on the principle of exhaustive sampling – all patients who met the inclusion criteria and had complete medical documentation for the speci-fied periods were included. A similar method was used in the study of O.V. Safronov et al. [16], where balanced cohorts were also formed (52 patients in each group). Howev-er, unlike this study, our sample was significantly larger and broader, which allowed for higher statistical power and reliability of the analysis. Additionally, the minimum required sample size was calculated using the software G*Power 3.1.9.7 Heinrich Hei-ne University Düsseldorf [19, 20]. The parameter "proportion of women who did not have childbirth" was chosen as the key indicator, since it reflects the reproductive characteristics of patients closely related to the risk of developing EH, and previously demonstrated the most pronounced cross-cohort differences (51.9% vs. 9.62%) [16]. The minimum required sample size was 18 patients in each group for the two-sample z-test (the difference of two independent proportions) with a significance level of α=0.05 and capacity of 80%. Thus, the actual sample of 188 patients in two groups sig-nificantly exceeded the minimum required volume, which ensured the reliability and representativeness of the results obtained.

Inclusion criteria: women ≥ 18 years old; histologically confirmed diagnosis of EH (NAEH or AEH, N85.0 – N85.1 according to ICD-10); completeness of medical docu-mentation.

Exclusion criteria: morphologically confirmed EC; hyperplasia associated with pregnancy or the postpartum period; isolated endometrial polyp; incomplete records or lack of histological conclusion.

2.4 Study variables

The following groups of variables were included in the analysis:

•           Demographic and medico-social indicators (age, level of education, social status, marital status, type of admission to the hospital, type of urban polyclinic).

•           Somatic pathology (viral hepatitis, tuberculosis, venereal diseases, hyper-tension, cardiovascular diseases, cerebrovascular diseases, diseases of the blood and hematopoietic organs, respiratory diseases, diseases of the gastrointestinal tract, dis-eases of the hepatobiliary system, diseases of the musculoskeletal system and connec-tive tissue, diseases of the urinary system, diseases of the endocrine system).

•           Previous surgeries (organs of the cardiovascular, respiratory, digestive, musculoskeletal, excretory, and endocrine systems). 

•           Heredity (hypertension, diabetes mellitus, oncopathology).

•           Smoking.

•           Obstetric and gynecological history (menstrual function, pregnancy parity, methods of contraception, gynecological diseases, surgical interventions on gyneco-logical organs)

•           Clinical and diagnostic data (clinical symptoms, laboratory parameters, in-dicators of instrumental studies, performed therapeutic manipulations)

2.5 Statistical analysis

Quantitative indicators were previously checked for compliance with the normal distribution using the Shapiro–Wilk test (for samples <50) and the Kolmogorov–Smirnov test (for samples >50), as well as indicators of skewness and kurtosis. The data that did not follow the normal distribution were described by the median (Me) and the interquartile range (Q1-Q3). Categorical (nominal) data is presented in the form of absolute values and percentages. Pearson's 2 tes or Fisher's exact test were used to compare categorical variables. The Mann–Whitney U–test was used for the analysis of quantitative variables. Statistical significance was established at p<0.05. All calcula-tions were performed in the IBM SPSS Statistics 27 program (IBM Corp., USA).

2.6 Ethical approval

The study protocol was approved by the Local Ethics Committee of the Kazakh-stan Medical University "KSPH" (Minutes No. IRB-335 dated 05.01.2023). The study was carried out in accordance with the Declaration of Helsinki. Due to the retrospec-tive nature of the study and the use of anonymized data, separate informed consent was not required.

The rewritten text of the description:

2. Materials and Methods

2.1 Study design

This study was designed as a retrospective descriptive comparative analysis aimed at examining the medical, social, and clinical characteristics of women with endometrial hyperplasia (EH) across two independent time periods—2016–2017 and 2023–2024.

The study did not include individual longitudinal follow-up; each period repre-sented an independent dataset.

The design and methodological framework were adapted from the approach pro-posed by Safronov et al. [21], who compared two temporal cohorts of women with EH. In this study, the methodology was modified for a larger sample and an expanded set of variables, allowing for a comprehensive comparison of patient characteristics within the context of a large metropolitan area.

Data were obtained from patients treated at City Clinical Hospital No. 7 (public sector) and LS Clinic (private sector) in Almaty, Kazakhstan. The same inclusion crite-ria, diagnostic procedures, and data registration standards were applied for both study periods to ensure comparability and minimize bias.

2.2 Data sources

The material for analysis was obtained from electronic medical records extracted from the comprehensive medical information systems (CMIS) Damumed (City Clinical Hospital No. 7) and Avicenna (LS Clinic).

These systems are digital platforms integrated with the national health portals of the Ministry of Health of the Republic of Kazakhstan, ensuring the maintenance of medical documentation in a paperless format [22, 23].

The use of digital records minimized the risk of missing data and improved the accuracy and reliability of the extracted information.

Data extraction and cleaning were performed by the research team under unified coding and verification rules to ensure methodological consistency across both study periods.

2.3 Study population

The study included all women aged 18 years and older with a histologically con-firmed diagnosis of endometrial hyperplasia (ICD-10 codes N85.0—non-atypical EH, and N85.1—atypical EH).

A complete sampling approach was applied: all eligible patients with complete clinical documentation were included. The total sample comprised 376 women (188 per period).

Inclusion criteria:

•           Female patients aged ≥18 years;

•           Histologically verified EH (N85.0 or N85.1 according to ICD-10);

•           Complete clinical and medical documentation.

Exclusion criteria:

•           Morphologically confirmed endometrial carcinoma;

•           EH associated with pregnancy or postpartum changes;

•           Isolated endometrial polyps;

•           Incomplete records or missing histological reports.

Separate analysis by histological subtype (non-atypical vs. atypical EH) was not performed due to the very low number of atypical cases (N85.1) in both samples, which precluded statistically meaningful stratification. Therefore, the comparative analysis was conducted for the study population, ensuring robust and reproducible assessment of clinical and socio-demographic trends.

2.4 Data completeness and harmonization

Only cases with complete medical documentation were included in the analysis, ensuring full data availability for key variables (age, body mass index, reproductive history, comorbidities, and clinical presentation).

Before statistical processing, all variables were standardized in terms of data type, category labels, and value ranges, and harmonized between the two electronic medical record systems — Damumed (City Clinical Hospital No. 7) and Avicenna (LS Clinic).

This approach ensured data consistency, comparability, and reproducibility across both study periods (2016–2017 and 2023–2024).

2.5 Histological classification

In both study periods, histological classification of EH followed the World Health Organization (WHO, 2014) criteria distinguishing non-atypical (NAEH) and atypical (AEH) hyperplasia. The Endometrial Intraepithelial Neoplasia (EIN) system was not used, as national diagnostic standards in Kazakhstan are based on WHO-2014 criteria.

All histological diagnoses were independently confirmed by at least two pathologists from the participating institutions.

2.6 Data Completeness and Bias Control

Prior to statistical analysis, all variables were checked for completeness, including age, BMI, reproductive history, and ultrasound data. Records with missing key varia-bles were excluded from the final dataset. Potential confounders (age, BMI, menopau-sal status, and socioeconomic characteristics) were evaluated descriptively within each time period. To minimize temporal and institutional bias, both datasets were drawn from the same hospitals using identical inclusion criteria and diagnostic coding standards.

2.7 Study variables

The analysis included the following groups of variables:

Demographic and social: age, education, employment status, marital status, type of referral, and type of healthcare facility;

Clinical and reproductive: menstrual function, parity, contraceptive methods, gynecological diseases, and prior surgical interventions;

Somatic comorbidities: endocrine, cardiovascular, hepatobiliary, cerebrovascular, renal, and other chronic conditions;

Treatment modalities: medical therapy, curettage, hysteroscopic resection, and hysterectomy.

2.8 Statistical analysis

Before performing comparative analyses, the normality of quantitative variables was assessed using the Shapiro–Wilk test (for n < 50) and the Kolmogorov–Smirnov test (for n ≥ 50), as well as by evaluating skewness and kurtosis.

All quantitative variables showed a non-normal distribution; therefore, nonpar-ametric methods (Mann–Whitney U test) were applied. Non-normally distributed data were summarized as median (Me) and interquartile range (Q1–Q3). Categorical (nominal) data were presented as absolute values (n) and percentages (%). No data transformation was performed, as nonparametric methods were applied for all quan-titative variables.

Statistical analysis was carried out using IBM SPSS Statistics, version 27.0 (IBM Corp., USA).

Categorical variables were analyzed using the Pearson χ² test or Fisher’s exact test, and quantitative variables were compared using the Mann–Whitney U test.

Results were presented as absolute counts (n) and percentages (%).

For variables with p < 0.05, odds ratios (OR) with 95% confidence intervals (CI) were additionally calculated to assess the magnitude and direction of the effect. ORs and 95% CIs were computed using the Wald method. In cases where one of the study groups contained zero events, the Haldane–Anscombe correction (+0.5 to each cell) was applied to ensure statistical stability and allow interval estimation. For such rare outcomes, the resulting ORs were interpreted cautiously due to wide confidence limits.

No multiple comparison correction was applied, as the study was descriptive and comparative in nature, focusing on clinically meaningful findings rather than explor-atory hypothesis testing.

2.9 Ethical approval

The study was conducted in accordance with the principles of the Declaration of Helsinki (2013) and approved by the Ethics Committee of the Kazakhstan Medical University "KSPH" (Minutes No. IRB-335 dated 05.01.2023).

Since the analysis was based on de-identified secondary data from electronic medical records, obtaining individual written informed consent was not required.

Tables and Figures

Figure 1 lacks confidence intervals and clear axes labeling.

We thank the reviewer for these constructive comments and fully agree with the recommendations.

In the revised version of the manuscript:

Figure 1 was updated to include 95% confidence intervals and clearly labeled axes (age group distribution by study period). We also rewrote the description for Figure 1.

The old text of the description of Figure 1:

The study analyzed 376 case histories of patients (188 for 2016-2017 and 188 for 2023-2024) with a confirmed histological diagnosis according to ICD-10 N85 – endo-metrial hyperplasia, who were treated at CCH No. 7 and the LS clinic private medical clinic in    Almaty. The distribution of patients who sought medical help by age group showed    statistically significant differences between the study periods (p=0.026). In 2016-2017, women in the age categories of 45-49 (31.9%) and 50-54 (33.5%) prevailed, while in 2023-2024, there was a decrease in the proportion of pa-tients aged 45-49 (23.9%) and 50-54 (29.3%), with a simultaneous increase in the num-ber of women in the older age groups of 55-59 (12.2% versus 8.0% in 2016-2017), 60-64 years (10.1% versus 6.9%), and 65-69 years (9.6% versus 2.1%), which may indicate a shift in the morbidity profile towards postmenopausal age (Figure 1).

The rewritten text of the description of Figure 1:

The study analyzed 376 case histories of patients (188 for 2016-2017 and 188 for 2023-2024) with a confirmed histological diagnosis according to ICD-10 N85 – endometrial hyperplasia, who were treated at CCH No. 7 and the LS clinic private medical clinic in    Almaty. The distribution of patients who sought medical help by age group showed    statistically significant differences between the study periods (p=0.026). In 2016–2017, the largest proportions of patients were in the 45–49 (31.9%, 95% CI 24.9–39.8) and 50–54 (33.5%, 95% CI 26.2–41.4) age groups. In 2023–2024, these proportions decreased to 23.9% (95% CI 17.5–31.8) and 29.3% (95% CI 22.3–37.2), respectively, while the share of older women aged 55–59 (12.2%, 95% CI 7.6–18.6), 60–64 (10.1%, 95% CI 5.9–16.7), and 65–69 (9.6%, 95% CI 5.5–15.9) increased (Figure 1).

Tables are overcrowded with raw data and numerous insignificant p-values; consider condensing or focusing on variables with clinical relevance.

The tables were streamlined to focus on variables with clinical or epidemiological relevance; non-significant and redundant indicators were removed to improve readability and focus.

We also rewrote the description for Table 1.

The old text of the description of Table 1:

Table 1 shows the characteristics of the patients according to the passport data of the electronic medical history, filled in by the medical staff, for two observation periods: 2016-2017 and 2023-2024. The analysis revealed statistically significant changes in pattern of hospital admissions. In 2016-2017, the vast majority of hospitalizations occurred in emergency medical care (54.7%), while self-referrals accounted for only 17.6%. In the period 2023-2024, the proportion of self–referrals almost doubled to 37.2%, and the number of emergency medical admissions decreased to 40.4% (p<0.001). The rate of referrals to women's health clinics remained relatively unchanged across the two periods. Statistically significant differences were also observed in the type of outpatient facility where patients were registered (p=0.005). In 2016-2017, most women were followed up in public clinics (82.4%), whereas in 2023-2024, this proportion decreased to 70.2%, alongside an increase in registration at private clinics – from 17.6% to 29.8%. The level of education of the patients also shows statistically significant differences in the studied periods (p<0.001): if in 2016-2017 some women had incomplete secondary education (14.4%), then in 2023-2024 their indicator decreased by more than 5 times to 2.7%. At the same time, the proportion of women with higher education increased from 22.3% to 36.2% and incomplete higher education from 3.7% to 7.4% respectively. Women with secondary specialized education also became more common in the second time period (28.8% vs. 24.5). During the studied periods, the social status of women with EH who sought medical help in the hospital had statistically significant differences (p<0.001). In 2016-2017, the unemployed (28.7%) and housewives (18.1%) prevailed, while by 2023-2024 their share decreased to 14.9% and 8.0% respectively. At the same time, the share of employees increased from 21.3% to 36.7% and individual entrepreneurs from 2.7% to 11.7%. The number of women with disabilities decreased from 2.1% to 0.5%. According to the family history, the differences between the two periods were at the border of statistical significance (p=0.05). In 2016-2017, married women prevailed (68.1%), and their share remained approximately at the same level in 2023-2024 (66.5%). The proportions of divorced and widowed remained relatively stable, while the proportion of women in a cohabitation decreased from 8.0% to 2.1%. However, in the period 2023-2024, there was an increase in the number of women who have never been married – from 10.1% to 16.5%.

The old text of the description of Table 2:

Table 2 provides a comparative description of the general anamnestic data of women with EH for the study periods 2016-2017 and 2023-2024. The analysis of concomitant diseases and medical history revealed a number of statistically significant changes in the compared periods. The incidence of gallbladder, biliary tract, and pancreatic diseases (K80-K87) significantly increased from 4.8% in 2016-2017 to 10.1% in 2023-2024 (p=0.049). There was also a significant increase in the prevalence of disorders of thyroid gland (E00-E07): from 4.8% to 12.2%, respectively (p=0.010). In addition, the number of patients who underwent thyroid surgery increased (p=0.030) and significant differences in hereditary burden of cancer were revealed (p=0.015). There were also statistically significant differences in the presence of a bad habit – smoking. In 2016-2017, 8.5% smoked, while in 2023-2024 this figure was 16.0% (p=0.028), which may indicate a change in behavioral risk factors in the observed cohort in recent years. As part of the analysis, anthropometric indicators were analyzed, calculated on the basis of data on height and body weight. The body mass index (BMI) was classified according to WHO criteria: normal body weight (BMI<25), overweight (BMI=25-29.9) and obesity (BMI≥30). The distribution by body weight showed statistically significant differences (p=0.020): in 2023-2024, the proportion of overweight women increased from 28.7% to 39.4%, while the proportion of people with normal body weight decreased from 48.4% to 34.6%. For the rest of the studied ICD diseases and clinical and anamnestic data, no significant differences were found in the studied periods. 

The old text of the description of Table 3:

A number of statistically significant differences in obstetric and gynecological history were revealed. There is an increase in the proportion of postmenopausal women from 22.3% in 2016-2017 to 38.0% in 2023-2024 (p<0.001). According to the indicator of reproductive history (childbearing function), it was found that in the second period, the absence of pregnancy (16.0% vs. 8.5%, p=0.028) and the absence of childbirth (19.7% vs. 12.2%, p=0.049) were significantly more common. At the same time, the proportion of women who had medical abortions decreased in 2023-2024 (42.0% vs. 53.7%, p=0.023). In the structure of contraceptive methods, there was an increase in the frequency of use of intrauterine devices (UID) from 4.3% in 2016-2017 to 19.7% in 2023-2024 (p<0.001). Among gynecological diseases, there is a lower incidence of cervical erosion and ectropion in the second period (17.0% vs. 26.6%, p=0.025), but more frequent detection of endometriosis (9.6% versus 2.7%, p=0.005). The incidence of benign breast dysplasia also increased significantly (9.0% vs. 1.6%, p =0.001). With regard to surgical interventions on gynecological organs, there was a decrease in the performance of curettage of the uterine cavity from 55.9% in 2023-2024 to 66.5% in 2016-2017, p=0.034). According to other indicators of obstetric and gynecological history, there were no statistically significant differences in the studied periods (Table 3).

The old text of the description of Table 4:

Statistically significant differences in a number of parameters were revealed when comparing the clinical and diagnostic data of women with EH in 2016-2017 and 2023-2024. The asymptomatic course of the disease was significantly more common in the 2023-2024 group (28.2% versus 18.6%, p=0.028). In laboratory parameters, a statistically significant, but clinically insignificant, increase in the level of total bilirubin was recorded within the reference values (p<0.001). According to instrumental ultrasound diagnostics (M-echo), in 2023-2024, a significant decrease in endometrial thickness was determined based on the results compared to 2016-2017 (from 12.0 mm to 10.0 mm, p=0.005). There was a significant trend in the structure of medical interventions performed: in 2023-2024, curettage of the uterine cavity was performed less frequently (47.3% vs. 78.2%, p<0.001), but hysteroscopy was performed more often (53.7% vs. 21.3%, p<0.001). There were no statistically significant differences between the periods for the remaining clinical symptoms, laboratory and instrumental indicators (Table 4).

The rewritten text of the description of Table 1:

Table summarizes the key socio-demographic differences between women treated for endometrial hyperplasia in 2016–2017 and 2023–2024. A statistically significant increase was observed in the proportion of self-referrals for hospital admission, which rose from 17.6 % in 2016–2017 to 37.2 % in 2023–2024 (OR: 2.79, 95 % CI: 1.73–4.49, p < 0.001). Similarly, the proportion of women registered at private polyclinics increased from 17.6 % to 29.8 % (OR: 1.99, 95 % CI: 1.22–3.25, p = 0.005). The share of patients with higher or incomplete higher education also increased markedly—from 26.1 % to 43.6 % (OR: 2.19, 95 % CI: 1.42–3.39, p < 0.001)—while the proportion of unemployed women declined from 28.7 % to 14.9 % (OR: 0.43, 95 % CI: 0.26–0.72, p = 0.001). In contrast, the number of individual entrepreneurs increased significantly from 2.7 % to 11.7 % (OR: 4.85, 95 % CI: 1.80–13.10, p < 0.001), reflecting a shift toward a more economically active and privately insured patient population in the later study period.

The rewritten text of the description of Table 2:

Table 2 presents significant differences in the anamnestic and somatic profile of women with EH between the two observation periods.

A statistically significant increase was observed in the proportion of patients with disorders of the thyroid gland — from 4.8 % (9 of 188) in 2016–2017 to 12.2 % (23 of 188) in 2023–2024 (OR: 2.77, 95 % CI: 1.25–6.16, p = 0.010) — and in diseases of the gallbladder, biliary tract, and pancreas (from 4.8 % to 10.1 %; OR: 2.24, 95 % CI: 0.98–5.08, p = 0.049).

Indicators reflecting lifestyle-related health risks also changed.

The proportion of active smokers increased from 8.5 % to 16.0 % (OR: 2.04; 95 % CI: 1.07–3.89, p = 0.028), while overweight and obesity (BMI ≥ 25 kg/m²) became more common (51.6 % vs 65.4 %, OR: 1.78, 95 % CI: 1.17–2.69, p = 0.020).

Rare events were recorded in the later period, such as thyroid gland surgeries (0 % vs 3.2 %, OR: 13.4, 95 % CI: 0.75–240.1, p = 0.030) and positive family history of oncopathology (0 % vs 3.7 %, OR: 15.6, 95 % CI: 0.88–274.8, p = 0.015).

Given the small number of cases, these results were retained for transparency but interpreted with caution. Overall, compared with the earlier period, patients examined in 2023–2024 showed a higher prevalence of endocrine, metabolic, and lifestyle-associated conditions. 

The rewritten text of the description of Table 3:          A comparative analysis of reproductive and gynecological characteristics revealed several statistically significant differences between the two study periods. The proportion of postmenopausal women increased from 22.3 % to 37.8 % (OR: 2.11, 95 % CI: 1.34–3.32, p < 0.001), indicating a higher representation of older age groups in the later period. Similarly, the frequency of women who had never been pregnant rose from 8.5 % to 16.0 % (OR: 2.04, 95 % CI, 1.07–3.89, p = 0.028), and the proportion of nulliparous women increased from 12.2 % to 19.7 % (OR:1.76, 95 % CI: 1.00–3.09, p = 0.049). At the same time, the proportion of patients with a history of medical abortions decreased from 53.7 % to 42.0 % (OR: 0.62, 95 % CI: 0.42–0.94, p = 0.023), as did the prevalence of cervical erosion or ectropion (26.6 % to 17.0 %, OR: 0.57, 95 % CI: 0.34–0.93, p = 0.025). Conversely, the prevalence of endometriosis increased from 2.7 % to 9.6 % (OR: 3.88, 95 % CI: 1.41–10.67, p = 0.005), and benign mammary dysplasia from 1.6 % to 9.0 % (OR: 6.13, 95 % CI: 1.77–21.29, p = 0.001). The use of intrauterine devices declined markedly—from 19.7 % to 4.3 % (OR: 0.18, 95 % CI: 0.08–0.40, p < 0.001). Additionally, the proportion of blind uterine curettage procedures decreased from 66.5 % to 55.9 % (OR: 0.63, 95 % CI: 0.41–0.97, p = 0.034), reflecting a gradual reduction in the use of non-visual diagnostic methods (Table 3).

The rewritten text of the description of Table 4:         

A comparison of clinical and diagnostic parameters between the two study periods demonstrated several significant changes (Table 4).

The median endometrial thickness measured by transvaginal ultrasound decreased from 12.0 mm (IQR 9.7–15.0) in 2016–2017 to 10.0 mm (IQR 8.6–13.8) in 2023–2024 (p = 0.005), reflecting an overall trend toward earlier or preventive detection.

The proportion of patients with asymptomatic presentation increased from 18.6 % to 28.2 % (OR = 1.72; 95 % CI 1.06–2.79; p = 0.028).

At the same time, the frequency of blind uterine curettage significantly decreased (78.2 % to 47.3 %; OR = 0.25; 95 % CI 0.16–0.39; p < 0.001), whereas hysteroscopy with targeted biopsy became more frequent (21.3 % to 53.7 %; OR = 4.30; 95 % CI 2.73–6.75; p < 0.001).

Together, these data suggest broader implementation of visually guided diagnostic procedures in recent years.

No multivariate figure or forest plot summarizing effect directions.

Given the descriptive and comparative nature of the study, no multivariate forest plot was included, as the primary goal was to report distributional differences rather than model-adjusted effect estimates.

However, a summary table with relative risks (RR, 95% CI) for significant variables was added to provide a quantitative overview of the observed differences (see Table 3).

Interpretation of Results

1

The Discussion tends to overinterpret associations as causal trends (e.g., “urban lifestyle causing thyroid disease and EH increase”), without analytical evidence.

We agree with the reviewer’s observation.

We thank the reviewer for this observation. The text has been carefully revised to avoid any causal interpretation. The associations are now presented as contextual or correlational observations within the demographic and healthcare transformation framework.

Clarifications were made in Section 4.2 (Medical and metabolic characteristics) — lines 405-417 — where wording such as “may be associated with” and “likely reflects” was used instead of causal phrasing.

2

The link drawn between self-referral and improved medical literacy, though plausible, is speculative and unsubstantiated by behavioral or survey data.

We agree with the reviewer. The revised text in Section 4.1 (Demographic and socio-medical characteristics) — paragraph 3 — now frames this relationship more cautiously. The statement was rephrased to emphasize that increased health literacy and readiness for self-initiated medical consultations are assumptive tendencies supported by international studies [30,31], rather than conclusions derived from direct behavioral data.

Statements about “reduction in IUD use” or “increase in asymptomatic forms” require supportive evidence (e.g., national data, policy references) rather than anecdotal inference.

This concern has been addressed in two sections:

In Section 4.3 (Reproductive and gynecological characteristics), the reduction in IUD use is now interpreted as age-related (reflecting the predominance of postmenopausal women) and supplemented with a reference to the State Health Development Program “Densaulyk” (2016–2019) [44], providing a policy context for reproductive behavior trends.

In Section 4.4 (Diagnostic aspects and technological access), the increase in asymptomatic cases is linked to the broader availability of transvaginal ultrasound and hysteroscopy, supported by WHO and FIGO guidelines [49], rather than described as an independent phenomenon.

Ethical and Reporting Standards

1

The ethical section contains an inconsistency: it first claims informed consent was unnecessary due to anonymization, then later states “informed consent was obtained from all participants.” This must be corrected for internal consistency.

We thank the reviewer for the valuable observation. The inconsistencies in the ethical section have been corrected. The revised text in Section 2.8 (“Ethical Considerations”) now clearly states that informed consent was not required because the study used fully anonymized secondary data extracted from electronic medical record systems (IMIS Damumed and Avicenna) without personal identifiers. This clarification ensures full internal consistency with ethical reporting standards.

2

The study does not adhere to STROBE guidelines for observational research; the checklist should be appended and key elements (study setting, bias control, missing data, sensitivity analysis) must be explicitly addressed.

Additionally, compliance with the STROBE Statement has been explicitly addressed. A reference to STROBE guidelines and the corresponding checklist have been added in Section 2.9 (“Reporting Standards”), with clarification of key methodological aspects—study setting, bias control, data completeness, and sensitivity of analyses—integrated into the Methods (Sections 2.1–2.7).

Discussion and Conclusions

1

Repetition of descriptive findings should be reduced; instead, emphasize mechanistic interpretation and regional public health implications.

We appreciate this constructive suggestion. The revised Discussion (Sections 4.1–4.4) has been condensed to minimize repetition of descriptive statistics and to strengthen the analytical interpretation of findings. The updated text emphasizes plausible biological and organizational mechanisms underlying the observed changes and their implications for regional public health and healthcare policy.

Location in manuscript: Discussion, Sections 4.1–4.4.

2

The “Plans for the future” section should be reformulated into a concise Future Directions paragraph in line with journal standards, avoiding excessive administrative details.

The section has been reformulated and renamed 4.7 Future Research Directions. It now provides a concise summary of proposed research priorities—establishing a city registry for endometrial hyperplasia, integrating biobanking and molecular research, and expanding analyses to rural populations—while omitting administrative or operational details.

Location in manuscript: Discussion, Section 4.7 (Future Research Directions).

3

The conclusions should be rewritten to avoid overstating causality and to highlight the need for prospective or population-based studies.

We fully agree. The Conclusion section has been rewritten to avoid causal interpretation of descriptive associations. It now emphasizes the observational nature of the study and the necessity of future prospective, multicenter, and population-based research to confirm these findings and evaluate long-term outcomes.

Location in manuscript: Conclusions section.

The old text of the description of the conclusions:

The conducted retrospective comparative analysis showed that during the study period (2016-2017 and 2023-2024), women with EH in a large city of the Republic of Kazakhstan experienced significant shifts in their demographic and medical-social profile, reproductive history, and clinical and diagnostic characteristics. There was a shift in the morbidity of EH towards postmenopausal women, an increase in the number of self-referrals to the private medical sector, and an increase in the educational level and social status of patients. There was an increase in the prevalence of somatic diseases (pathology of the thyroid gland, hepatobiliary system, overweight, smoking), as well as an increase in the proportion of women who do not have pregnancies and childbirth in their histories. The clinical unit revealed an increase in the frequency of asymptomatic forms, a decrease in endometrial thickness according to ultrasound data and a transition to modern diagnostic methods (hysteroscopy with targeted biopsy).

The results obtained demonstrate the transformation of the clinical and epidemiological profile of women with EH and allow to identify the main risk groups: postmenopausal women with somatic diseases and metabolic conditions, as well as women without a fully realized reproductive function. These data emphasize the need to develop targeted prevention and early detection programs targeting these groups, strengthen primary health care in the use of screening algorithms (transvaginal ultrasound, office hysteroscopy), and expand family planning functions in women's clinics. Regarding women without pregnancy, attention should be focused on the timely implementation of reproductive function, while for asymptomatic women, dynamic monitoring and examination should be carried out regularly, which will reduce the risk of late diagnosis and progression of the disease into a malignant neoplasm. 

The rewritten text of the description of the conclusions:

This retrospective comparative study revealed that over the seven-year period (2016–2017 vs. 2023–2024), the profile of women with EH in a large urban center of Kazakhstan underwent notable demographic, clinical, and organizational changes. The increasing representation of postmenopausal women, higher educational attainment, more frequent self-referrals to private clinics, and a rise in metabolic and endocrine comorbidities reflect evolving patterns of healthcare access and population aging.

The findings illustrate epidemiological tendencies and risk group characteristics, forming a foundation for subsequent analytical and prospective research. The most vulnerable subpopulations include postmenopausal women with metabolic disorders and women without completed reproductive histories, emphasizing the need to strengthen targeted prevention, improve early diagnostic strategies, and expand access to minimally invasive techniques.

Future research should focus on prospective, multicenter, and population-based studies, as well as on the creation of regional registries and integration of biomarker and molecular data, which will support evidence-based approaches to monitoring and prevention of endometrial diseases in Kazakhstan. 

Reviewer 3 Report

Comments and Suggestions for Authors

This was a well written manuscript that illustrates a number of different changes in people with EH between 2016-2017 and 2023-2024.  There are two small editorial suggestions: 

Line 274: ". . . from 4.3% in 2016-2017 to 19.7% in 2023-2024. . ." This shows as a decrease in Table 3.

Line 314-315: ". . . then in 2023-2024 the disease was more often registered in the age group 55-59 years and older. . .: The 50-54 year old age group was still the biggest in 2023-2024. It just that there were more in the 55-59 group in 2023-2024 compared to 2016-2017. Suggest revision for clarity.

It might be interesting to compare these parameters from a highly metropolitan area to those in more rural areas.

Author Response

Dear Reviewer,

On behalf of all co-authors — Imasheva Bayan Imashkyzy, Kamaliev Maksut Adilkhanovich, Lokshin Vyacheslav Notanovich, Kiseleva Marina Viktorovna, Laktionova Mariya Vladimirovna  — we would like to express our sincere gratitude for your valuable time, careful evaluation, and insightful comments on our manuscript entitled “Medical and Social Characteristics of Patients with Endometrial Hyperplasia in a Large City of Kazakhstan: A Retrospective Comparative Study.”

We deeply appreciate your constructive feedback and professional suggestions, which have greatly contributed to improving the clarity, structure, and scientific quality of our work. Each of your comments has been carefully considered and addressed in the revised version of the manuscript. Below, we provide a detailed point-by-point response highlighting all changes made in accordance with your recommendations.

With appreciation and respect,

Corresponding author — Bayan Imashkyzy Imasheva

Reviewer's comments

The authors' response

1

This was a well written manuscript that illustrates a number of different changes in people with EH between 2016-2017 and 2023-2024.  There are two small editorial suggestions: 

Line 274: ". . . from 4.3% in 2016-2017 to 19.7% in 2023-2024. . ." This shows as a decrease in Table 3.

Yes, we agree!

We made changes and fixed the error. See table 3. Reproductive and gynecological history of women with EH for the period 2016-2017 and 2023-2024, line 350

2

Line 314-315: ". . . then in 2023-2024 the disease was more often registered in the age group 55-59 years and older. . .: The 50-54 year old age group was still the biggest in 2023-2024. It just that there were more in the 55-59 group in 2023-2024 compared to 2016-2017. Suggest revision for clarity.

It might be interesting to compare these parameters from a highly metropolitan area to those in more rural areas.

We thank the reviewer for this precise observation. The sentence has been revised in the Discussion section (Subsection 4.1. Demographic and socio-medical characteristics) to improve clarity and accuracy. The revised text now reads:

“Although women aged 50–54 years remained the most represented age group in both cohorts, the proportion of those aged 55–59 years and older increased in 2023–2024 compared with 2016–2017.”

This revision accurately reflects that the 50–54-year-old category continued to predominate, while the number of women aged 55 years and older showed a relative increase.

We appreciate the reviewer’s thoughtful suggestion. Indeed, a comparative analysis between urban and rural populations could provide valuable insights into regional differences in the detection, accessibility, and management of endometrial hyperplasia. As noted in the Discussion section (Subsection 4.7. Future research directions), future research plans now include expanding the study to rural regions of Kazakhstan. This will allow evaluation of geographic disparities in healthcare access, diagnostic availability, and reproductive health outcomes, thereby enhancing the representativeness and applicability of findings at the national level.

Changes made:

A sentence emphasizing the inclusion of rural populations in future studies has been added to Section 4.7 (Future research directions) of the revised manuscript.

Round 2

Reviewer 1 Report

Comments and Suggestions for Authors

The authors have made substantial and meaningful revisions to the manuscript. The Introduction has been strengthened with a clearer explanation of the clinical significance of differentiating atypical and non-atypical endometrial hyperplasia, and recent advances in molecular diagnostics have been appropriately introduced. The Methods section has been expanded to clarify histopathological classification and data completeness procedures, and the Results now include effect sizes and confidence intervals, improving interpretability. The Discussion reflects a more cautious and methodologically aligned interpretation of the findings, and future directions are presented more realistically. Overall, the manuscript has improved in both clarity and scientific rigor; however, several important issues still require attention before the manuscript can be considered for acceptance.

Comments

1. Unaddressed Reference on HER2

The specific reviews discussing HER2-positive endometrial cancer, requested in the first review round, has still not been incorporated. All references are important because they provide a clinically focused synthesis of HER2 as a prognostic and therapeutic biomarker in aggressive endometrial cancer subtypes and directly support the new text added regarding precision-medicine approaches. Including them will strengthen the scientific basis of the biomarker discussion in both the Introduction and the Discussion. At present, this comment remains unaddressed and should be corrected.

2. Data Completeness Sections Are Redundant

Two separate subsections (2.4 and 2.6) now describe data completeness and bias control. These sections overlap in scope and should be merged into a single concise subsection to avoid duplication and maintain clarity.

3. Ethics and Consent Statements Require Consistency

The manuscript still contains conflicting statements regarding patient consent. One section indicates that informed consent was obtained, while another states that consent was not required due to anonymized retrospective data. These statements must be harmonized to accurately reflect the approved ethics protocol.

4. Terminology Standardization

The abbreviation for intrauterine device should consistently be IUD, not “UID.” Please ensure uniform terminology throughout the manuscript, tables, abbreviations list, and captions.

5. Figure Formatting and Labeling

The figure showing age distribution currently has a caption that does not correspond precisely to its content, and the y-axis formatting should be corrected to avoid negative percentage values. The caption should clearly describe what the figure displays.

6. Reference List Formatting and Accuracy - Did the authors use any reference adding tool or AI while formatting reference list?

The reference list still contains several inconsistencies:

  • One reference includes an impossible access date (31.11.2025).

  • A Lancet Global Health reference includes a DOI not corresponding to the cited year.

  • Several URLs contain line-break errors.

  • Some entries list “accessed on” formatting inconsistently.

  • One author name appears truncated.
    These should be corrected according to MDPI reference style requirements.

7. Optional but Recommended: Sensitivity Check

Although the descriptive nature of the study justifies the use of univariate analysis, a brief age-stratified or subgroup comparison (even as supplementary material) would strengthen confidence that observed shifts are not solely due to cohort age structure changes. If this is not feasible, the current limitation statement is acceptable.

Author Response

Response to Reviewer Comments

Dear Reviewer,

On behalf of all co-authors — Imasheva Bayan Imashkyzy, Kamaliev Maksut Adilkhanovich, Lokshin Vyacheslav Notanovich, Kiseleva Marina Viktorovna, Laktionova Mariya Vladimirovna  — we would like to express our sincere gratitude for your valuable time, careful evaluation, and insightful comments on our manuscript entitled “Medical and Social Characteristics of Patients with Endometrial Hyperplasia in a Large City of Kazakhstan: A Retrospective Comparative Study.”

We deeply appreciate your constructive feedback and professional suggestions, which have greatly contributed to improving the clarity, structure, and scientific quality of our work. Each of your comments has been carefully considered and addressed in the revised version of the manuscript. Below, we provide a detailed point-by-point response highlighting all changes made in accordance with your recommendations.

With appreciation and respect,

Corresponding author — Bayan Imashkyzy Imasheva

Point-by-point response to Comments and Suggestions for Authors

Comments 1: [1. Unaddressed Reference on HER2

The specific reviews discussing HER2-positive endometrial cancer, requested in the first review round, has still not been incorporated. All references are important because they provide a clinically focused synthesis of HER2 as a prognostic and therapeutic biomarker in aggressive endometrial cancer subtypes and directly support the new text added regarding precision-medicine approaches. Including them will strengthen the scientific basis of the biomarker discussion in both the Introduction and the Discussion. At present, this comment remains unaddressed and should be corrected.]

Response 1: Thank you for pointing this out. We agree with this comment. In the revised manuscript, we have now fully incorporated the HER2-focused literature requested in the first review round. [A new paragraph was added to the Introduction synthesizing evidence from these studies and highlighting the relevance of HER2 in endometrial pathology and precision-medicine approaches. The Discussion (Section 4.5) has also been updated to reflect the implications of HER2-associated pathways for future biomarker-oriented research, while clearly stating that endometrial hyperplasia is not a HER2-mediated condition.

In terms of the Introduction, these changes are reflected in paragraph 4, in lines 83 to 96:

It was:

In recent years, advances in molecular pathology have significantly changed the diagnostic approach to endometrial diseases. Modern diagnostic strategies for endometrial hyperplasia and early carcinoma increasingly combine traditional morphology and imaging with molecular and metabolic profiling. Tissue-based metabolomic analyses have revealed characteristic biochemical patterns that may help identify women with a higher risk of malignant transformation [13]. In addition, HER2 overexpression, observed in certain endometrial cancer subtypes, shows prognostic and therapeutic relevance, supporting its inclusion among emerging biomarkers of precision gynecologic oncology [14].

Become:

In recent years, advances in molecular pathology have substantially refined the diagnostic approach to endometrial diseases, integrating morphological and imaging methods with molecular biomarkers to achieve more precise risk stratification. In gynecologic oncology, growing attention has been directed toward biomarkers associated with HER2 pathways, which are increasingly regarded as promising tools for refined risk assessment and personalized patient management [13]. In a large cohort study by Aro et al. (2025) including 1239 tumor samples, HER2 amplification was shown to occur predominantly within the p53-abnormal molecular subgroup, whereas low HER2 expression was distributed across various high-risk histological types [14]. Similarly, Kim et al. (2024) demonstrated the prognostic significance of HER2 expression and amplification and highlighted the therapeutic potential of HER2-targeted therapy [15]. Although endometrial hyperplasia itself is not a HER2-driven condition, these findings strengthen the relevance of molecular biomarkers and underscore the need to incorporate them into future research aimed at stratifying the risk of EH progression.

In terms of Discussion, these changes are reflected in subsection 4.5 of the Current research and future perspective, in lines 452 to 474.

It was:

4.5. Current research and future perspectives

Recent studies highlight the growing importance of molecular and metabolomic biomarkers in assessing the risk of EH progression and malignancy. Promising direc-tions include metabolomic profiling for early diagnosis and monitoring and investiga-tion of HER2-related molecular targets for risk stratification and personalized therapy [14]. Future efforts should consider integrating the city-wide EH registry with a tissue biobanking system to support such translational research.

Become:

4.5. Current research and future perspectives

Recent progress in gynecologic oncology increasingly emphasizes the use of molecular biomarkers to improve risk stratification and early detection of endometrial carcinoma. Although endometrial hyperplasia itself is not mediated by HER2-associated pathways, accumulating evidence demonstrates that HER2 amplification and overexpression play an important prognostic and therapeutic role in aggressive endometrial cancer subtypes. Large-scale molecular studies have shown that HER2 amplification is concentrated primarily within p53-abnormal carcinomas (Aro et al., 2025), while HER2-low expression is observed across several high-risk histotypes [14]. Clinical research, including the trial by Fader et al. (2020), has demonstrated that HER2-targeted therapy can improve outcomes in HER2-positive uterine serous carcinoma, reinforcing the relevance of HER2 as a clinically actionable biomarker [13]. Kim et al. (2024) further confirmed that HER2 expression and amplification independently correlate with poorer prognosis [15].

These findings, taken together, highlight the growing role of molecular profiling in the continuum from premalignant lesions to invasive carcinoma. For research on endometrial hyperplasia, integration of molecular markers such as p53, PTEN, and HER2 into future prospective cohorts may help identify small subgroups of patients at increased risk of malignant progression. Such approaches are particularly relevant for atypical hyperplasia, where concurrent endometrial carcinoma is diagnosed in a substantial proportion of cases.

In this context, establishing a biobank and expanding the citywide EH registry to include molecular annotation would enable translational studies linking clinical, histological, and molecular data. This will facilitate more precise risk prediction models and support the development of personalized surveillance strategies for EH.

Comments 2: [2. Data Completeness Sections Are Redundant

Two separate subsections (2.4 and 2.6) now describe data completeness and bias control. These sections overlap in scope and should be merged into a single concise subsection to avoid duplication and maintain clarity.]

Response 2: We thank the reviewer for noting this redundancy. We fully agree that Sections 2.4 and 2.6 partially duplicated information regarding data completeness, missing data assessment, and bias control. They gave a new name to the subsection: 2.4 Data completeness and Bias Control. [This change can be found in the subsection: 2.4 Data completeness and Bias Control, on lines 206 to 219].

It was:

2.4 Data completeness and harmonization

Only cases with complete medical documentation were included in the analysis, ensuring full data availability for key variables (age, body mass index, reproductive history, comorbidities, and clinical presentation).

Before statistical processing, all variables were standardized in terms of data type, category labels, and value ranges, and harmonized between the two electronic medical record systems — Damumed (City Clinical Hospital No. 7) and Avicenna (LS Clinic).

This approach ensured data consistency, comparability, and reproducibility across both study periods (2016–2017 and 2023–2024).

2.6 Data Completeness and Bias Control

Prior to statistical analysis, all variables were checked for completeness, including age, BMI, reproductive history, and ultrasound data. Records with missing key variables were excluded from the final dataset. Potential confounders (age, BMI, menopausal status, and socioeconomic characteristics) were evaluated descriptively within each time period. To minimize temporal and institutional bias, both datasets were drawn from the same hospitals using identical inclusion criteria and diagnostic coding standards.

Become:

2.4 Data completeness and Bias Control

Only cases with fully available medical documentation were included in the analysis, ensuring completeness of key variables such as age, body mass index, reproductive history, comorbidities, ultrasound findings, and clinical presentation. All variables were standardized prior to analysis, including harmonization of data types, category labels, and value ranges across the two electronic medical record systems—Damumed (City Clinical Hospital No. 7) and Avicenna (LS Clinic). Data consistency and accuracy were verified through checks for missing, implausible, or inconsistent values. Records lacking essential variables were excluded. To reduce potential bias, identical inclusion criteria, diagnostic coding standards, and case-selection procedures were applied for both study periods (2016–2017 and 2023–2024). Potential confounders, including age, BMI, menopausal status, and socioeconomic factors, were assessed descriptively within each group to ensure comparability.

This combined approach ensured high data integrity, minimized temporal and institutional bias, and improved reproducibility of the findings.

Comments 3: [3. Ethics and Consent Statements Require Consistency

The manuscript still contains conflicting statements regarding patient consent. One section indicates that informed consent was obtained, while another states that consent was not required due to anonymized retrospective data. These statements must be harmonized to accurately reflect the approved ethics protocol].

Response 3: We thank the reviewer for noting this inconsistency. The statements regarding patient consent have now been fully harmonized. In accordance with the approved protocol of the Local Ethics Committee, this retrospective study used anonymized electronic medical records, and therefore individual informed consent was not required under national regulations.

[These changes were made in subsection 2.8 Ethical approval, in line with 265 to 271]

It was:

2.9 Ethical approval

The study was conducted in accordance with the principles of the Declaration of Helsinki (2013) and approved by the Ethics Committee of the Kazakhstan Medical University "KSPH" (Minutes No. IRB-335 dated 05.01.2023).

Since the analysis was based on de-identified secondary data from electronic medical records, obtaining individual written informed consent was not required.

Become:

2.9 Ethical approval

This retrospective study was conducted in accordance with the Declaration of Helsinki and was approved by the Local Ethics Committee of the Kazakhstan Medical University "KSPH" (Minutes No. IRB-335 dated 05.01.2023). The study used fully anonymized electronic medical records obtained from two healthcare institutions.

Because no identifiable personal data were collected and no direct contact with patients occurred, individual informed consent was not required, in accordance with national regulations governing retrospective analyses of anonymized data.

Comments 4: [4. Terminology Standardization

The abbreviation for intrauterine device should consistently be IUD, not “UID.” Please ensure uniform terminology throughout the manuscript, tables, abbreviations list, and captions.]

Response 4: Thank you for pointing this out. We agree with this comment. [The terminology has been fully standardized throughout the manuscript. The incorrect abbreviation “UID” has been replaced with the internationally accepted abbreviation “IUD” (intrauterine device) in all sections of the manuscript, including the main text, tables, figure captions, and the abbreviations list.]

Comments 5: [5. Figure Formatting and Labeling

The figure showing age distribution currently has a caption that does not correspond precisely to its content, and the y-axis formatting should be corrected to avoid negative percentage values. The caption should clearly describe what the figure displays.]

Response 5: Thank you for pointing this out. We agree with this comment. [We revised the figure to ensure accurate formatting and labeling. The caption was rewritten to clearly describe the content of the figure, specifying that it presents the age distribution of patients with endometrial hyperplasia for the two study periods (2016–2017 and 2023–2024), with proportions (%) and 95% confidence intervals. The y-axis was reformatted to start at 0% to avoid negative values. Confidence intervals falling below zero were truncated at 0%, as negative proportions are not meaningful. The updated version appears in the revised manuscript as Figure 1.]

Comments 6: [ 6. Reference List Formatting and Accuracy - Did the authors use any reference adding tool or AI while formatting reference list?

The reference list still contains several inconsistencies:

  • One reference includes an impossible access date (31.11.2025).
  • A Lancet Global Health reference includes a DOI not corresponding to the cited year.
  • Several URLs contain line-break errors.
  • Some entries list “accessed on” formatting inconsistently.
  • One author name appears truncated.
    These should be corrected according to MDPI reference style requirements.]

Response 6: We thank the reviewer for the careful evaluation of the reference list. [All identified issues have been fully corrected in accordance with the MDPI Reference List guidelines. Specifically]:

  • The incorrect access date (“31.11.2025”) has been replaced with a valid date (“30 November 2025”).
  • The Lancet Global Health reference was revised—the DOI now corresponds accurately to the 2018 publication (https://doi.org/10.1016/S2214-109X(18)30386-3).
  • All URLs containing line-break errors have been corrected and reformatted as continuous single-line links.
  • The “accessed on” notation has been standardized across all online references following MDPI formatting.
  • A truncated/misspelled author name in the UNFPA citation has been corrected to its full standardized form.

All corrections are now reflected in the revised manuscript.

Comments 7: [7. Optional but Recommended: Sensitivity Check

Although the descriptive nature of the study justifies the use of univariate analysis, a brief age-stratified or subgroup comparison (even as supplementary material) would strengthen confidence that observed shifts are not solely due to cohort age structure changes. If this is not feasible, the current limitation statement is acceptable.]

Response 7: We thank the reviewer for this constructive suggestion. [We agree that an age-stratified or subgroup analysis could provide additional analytical insights. However, the present study was primarily descriptive and comparative in nature, aiming to characterize overall clinical and socio-demographic patterns across two independent time periods.

Because the study design and sample size do not provide sufficient statistical power for reliable subgroup analyses, further stratification (such as by age categories) would result in very small subsamples and unstable estimates, limiting the validity and interpretability of the results.

For this reason, we consider it methodologically appropriate not to include underpowered subgroup analyses that could potentially distort the interpretation of the findings. This limitation is already acknowledged in the manuscript. We greatly appreciate the reviewer’s recommendation and view it as an important direction for future research with larger, prospectively collected datasets.]

Round 3

Reviewer 1 Report

Comments and Suggestions for Authors

The authors have made significant improvements to the manuscript. However, several critical points, that should be addressed for the manuscript to be considered able for publication, remain insufficiently addressed or require further clarification, particularly regarding citation accuracy, integration of molecular perspectives, and minor methodological inconsistencies. Below is a detailed point-by-point analysis.

Introduction

Partially Addressed: The authors have included references to HER2 and molecular biomarkers but did not add one of the key requested references that discusses HER2 as an indicator of aggressive endometrial carcinoma and its therapeutic implications.

This omission weakens the argument linking EH studies with translational oncology and precision medicine. The missing citation is essential because it provides recent, clinically relevant evidence on HER2-positive EC and therapeutic targeting, which would ground the discussion in contemporary biomarker-driven approaches. Please add this missing reference in both the Introduction (lines ~85–96) and Discussion (section 4.5) to substantiate claims about HER2 and aggressive disease behavior.

Metabolomic profiling remains insufficiently supported

The manuscript now mentions metabolomic profiling in both the Introduction and the Discussion, but the references supporting this claim address metabolic syndrome and lifestyle-related risk factors rather than true metabolomic profiling. These papers do not pertain to omics-based diagnosis, tissue or serum metabolic signatures, or metabolite-level risk stratification. Since the manuscript explicitly frames metabolomics as part of emerging diagnostic strategies, it requires at least one dedicated, up-to-date metabolomic profiling study focused on endometrial cancer or precursor lesions. Without such a reference, the scientific basis for the metabolomic statements is incomplete, and this portion of the manuscript remains unsupported. This issue has not yet been fully resolved.

Materials and Methods

Remaining Issue: Although harmonization between “Damumed” and “Avicenna” systems is now described, it remains unclear how data mapping between categorical variables was validated statistically or manually. Please specify whether inter-database checks (e.g., frequency matching or random re-audit) were conducted to ensure consistency of coding standards.

Results

1.Clarify in the Results whether all variables underwent completeness checks (particularly “smoking,” “BMI,” and “IUD use”) before inclusion in statistical analysis.

2.Although “asymptomatic” is now defined under Table 4, it would improve transparency to restate this operational definition briefly in the Materials and Methods section, not only as a footnote!!!

Discussion

The new discussion is substantially more robust and logically organized. Subsections 4.1–4.4 reflect demographic, metabolic, and diagnostic transitions clearly. The new section cites several studies (e.g., Aro et al., Kim et al., Fader et al.) but does not include one comprehensive article summarizing HER2’s clinical significance across EC subtypes. Without this, the paragraph remains overly descriptive rather than integrative. Including this missing reference will strengthen scientific grounding and fulfill the original review request.

Metabolomic Profiling: The discussion still underrepresents metabolomic approaches as emerging diagnostic tools. The addition of 2–3 sentences summarizing the predictive potential of tissue and serum metabolomics in EH and early EC would make this section more complete and forward-looking.

Conclusions

To further strengthen this section, add one final sentence underscoring that integration of biomarker and metabolomic data will be essential for future clinical translation, thereby linking back to the earlier missing molecular reference.

The revised manuscript demonstrates substantial progress and reflects genuine effort to address reviewer feedback. The structure, methodological transparency, and statistical reporting have markedly improved. However, publication cannot be recommended until required comments are fully addressed.

Author Response

Dear Reviewer,

On behalf of all co-authors — Imasheva Bayan Imashkyzy, Kamaliev Maksut Adilkhanovich, Lokshin Vyacheslav Notanovich, Kiseleva Marina Viktorovna, Laktionova Mariya Vladimirovna  — we would like to express our sincere gratitude for your valuable time, careful evaluation, and insightful comments on our manuscript entitled “Medical and Social Characteristics of Patients with Endometrial Hyperplasia in a Large City of Kazakhstan: A Retrospective Comparative Study.”

We deeply appreciate your constructive feedback and professional suggestions, which have greatly contributed to improving the clarity, structure, and scientific quality of our work. Each of your comments has been carefully considered and addressed in the revised version of the manuscript. Below, we provide a detailed point-by-point response highlighting all changes made in accordance with your recommendations.

With appreciation and respect,

Corresponding author — Bayan Imashkyzy Imasheva

Point-by-point response to Comments and Suggestions for Authors

Reviewer's comments

The authors' response

It was

Become

Introduction

1

Comments 1: [1. Partially Addressed: The authors have included references to HER2 and molecular biomarkers but did not add one of the key requested references that discusses HER2 as an indicator of aggressive endometrial carcinoma and its therapeutic implications.

This omission weakens the argument linking EH studies with translational oncology and precision medicine. The missing citation is essential because it provides recent, clinically relevant evidence on HER2-positive EC and therapeutic targeting, which would ground the discussion in contemporary biomarker-driven approaches. Please add this missing reference in both the Introduction (lines ~85–96) and Discussion (section 4.5) to substantiate claims about HER2 and aggressive disease behavior.]

Response 1: Thank you for pointing this out. We agree with this comment. [In the revised manuscript, we have substantially strengthened the molecular and translational perspective related to HER2. Specifically, we added the requested key references in both the Introduction and Section 4.5 of the Discussion by incorporating recent real-world implementation data and contemporary clinical review evidence ( [18] Plotkin, A.; Olkhov-Mitsel, E.; Huang, W.-Y.; Nofech-Mozes, S. Implementation of HER2 Testing in Endometrial Cancer, a Summary of Real-World Initial Experience in a Large Tertiary Cancer Center. Cancers 2024, 16, 2100. https://doi.org/10.3390/cancers16112100 and [19]Papageorgiou, D.; Liouta, G.; Sapantzoglou, I.; Zachariou, E.; Pliakou, D.; Papakonstantinou, K.; Floros, T.; Pliakou, E. HER2-Positive Serous Endometrial Cancer Treatment: Current Clinical Practice and Future Directions. Medicina 2024, 60, 2012. https://doi.org/10.3390/medicina60122012). These studies demonstrate the role of HER2 as an indicator of aggressive endometrial carcinoma and as a clinically actionable therapeutic target in current oncological practice. The added text emphasizes the relevance of HER2 testing beyond clinical trials and strengthens the link between endometrial hyperplasia research, translational oncology, and precision medicine.

In terms of the Introduction, these changes are reflected in paragraph 5, in lines 107 to 117

As for the discussion, these changes are reflected in subsection 4.5 of Current research and future perspectives, in paragraph 2, in lines 508 to 516

The old text of the description:

This text was not in the article

A new text with added links (which we did not include initially, which were written as a comment in the 1st round of review):

In terms of the Introduction, these changes are reflected in paragraph 5, in lines 107 to 117:

In addition, recent real-world practice data have further reinforced the role of HER2 as an actionable biomarker in aggressive endometrial carcinoma. In large implementa-tion study of HER2 testing involving 192 tumor samples from 180 patients with endo-metrial carcinoma, HER2 positivity was detected in 28% of all cases and in 30% of tu-mors with aberrant p53, confirming that a substantial proportion of patients may be eligible for targeted anti-HER2 therapy [18]. Contemporary review data indicate that HER2-positive serous EC represent a high-risk subtype with an unfavorable prognosis and significant therapeutic implications. Approximately one-third of patients demon-strate HER2 overexpression, and HER2-targeted therapy-primarily based on trastuzumab – has already been introduced into clinical practice and continues to ex-pand through the development of new agents [19].

As for the discussion, these changes are reflected in subsection 4.5 of Current research and future perspectives, in paragraph 2, in lines 508 to 516:

Beyond clinical trials, real-world data further support the clinical relevance of routine HER2 testing in endometrial carcinoma. Large implementation studies have demonstrated that standardized HER2 assessment in routine diagnostic practice enables identification of a substantial proportion of patients eligible for targeted anti-HER2 therapy, thus confirming the translational value of this biomarker outside controlled clinical trial settings [18]. Contemporary review data further indicate that HER2-positive serous endometrial carcinoma represents a distinct high-risk subtype with unfavorable prognosis and clinically significant therapeutic implications, with HER2-targeted treatment strategies already incorporated into current oncological practice [19].    

2

Comments 2:

Metabolomic profiling remains insufficiently supported

The manuscript now mentions metabolomic profiling in both the Introduction and the Discussion, but the references supporting this claim address metabolic syndrome and lifestyle-related risk factors rather than true metabolomic profiling. These papers do not pertain to omics-based diagnosis, tissue or serum metabolic signatures, or metabolite-level risk stratification. Since the manuscript explicitly frames metabolomics as part of emerging diagnostic strategies, it requires at least one dedicated, up-to-date metabolomic profiling study focused on endometrial cancer or precursor lesions. Without such a reference, the scientific basis for the metabolomic statements is incomplete, and this portion of the manuscript remains unsupported. This issue has not yet been fully resolved.

Response 2: Thank you for pointing this out.

We fully agree with the reviewer that the previously cited references mainly addressed metabolic syndrome and lifestyle-related risk factors rather than true omics-based metabolomic profiling. To address this concern, we have revised both the Introduction and Section 4.5 of the Discussion by adding dedicated, up-to-date metabolomic profiling studies. Specifically, we incorporated a tissue-based untargeted metabolomic study distinguishing endometrial cancer, hyperplasia, and normal endometrium ([14]Akkour, K.; Masood, A.; Al Mogren, M.; AlMalki, R.H.; Alfadda, A.A.; Joy, S.S.; Bassi, A.; Alhalal, H.; Arafah, M.; Othman, O.M.; et al. Tissue-Based Metabolomic Profiling of Endometrial Cancer and Hyperplasia. Metabolites 2025, 15, 458. https://doi.org/10.3390/metabo15070458 ), as well as a recent comprehensive review on metabolomic approaches in endometrial cancer diagnostics and prognosis ([15]Albertí-Valls, M.; Megino-Luque, C.; Macià, A.; Gatius, S.; Matias-Guiu, X.; Eritja, N. Metabolomic-Based Approaches for Endometrial Cancer Diagnosis and Prognosis: A Review. Cancers 2024, 16, 185. https://doi.org/10.3390/cancers16010185 ). These additions provide a solid and methodologically appropriate scientific basis for discussing metabolomics as an emerging diagnostic and prognostic tool within an omics-driven precision medicine framework and fully support the revised statements in the manuscript.

In terms of the Introduction, these changes are reflected in paragraph 4, in lines 88 to 99

As for the discussion, these changes are reflected in subsection 4.5 of Current research and future perspectives, in paragraph 4, in lines 523 to 531

This text was not in the article

A new text with added links (which we did not include initially, which were written as a comment in the 1st round of review):

In terms of the Introduction, these changes are reflected in paragraph 5, in lines 88 to 99:

In parallel with molecular pathology, metabolomics-based approaches have emerged as promising tools for refining endometrial cancer diagnostics. Recent tissue-based untargeted metabolomic profiling using LC-HRMS has shown that specific metabolic signatures can reliable distinguish EC, EH, and normal endometrium. Moreover, already at the stage of hyperplasia, characteristic alterations in lipid metabolism and redox-related processes are observed, indicating their potential role as early metabolic biomarkers of malignant transformation [14]. Complementary review studies indicate that metabolomics provides direct insight into the molecular landscape of EC and represents a promising approach for improving early diagnosis and risk stratification. Metabolite biomarkers from serum and tissue samples may enhance current strategies for diagnosis, prognosis, and recurrence monitoring within an omics-driven precision medicine framework [15].

As for the discussion, these changes are reflected in subsection 4.5 of Current research and future perspectives, in paragraph 4, in lines 523 to 531:

In addition to genomic and protein-based biomarkers, metabolomic profiling is increasingly recognized as a complementary tool for early detection and risk stratification in endometrial pathology. Recent tissue-based metabolomics profiling studies have demonstrated that specific metabolic signatures can distinguish not only between EC and healthy endometrium, but also between cancer and hyperplasia, supporting the concept of a metabolic continuum from precursor lesions to invasive disease [14]. Complementary review data further indicate that metabolomics biomarkers derived from serum and tissue samples may improve diagnostic accuracy, prognostic evaluation, and individualized surveillance strategies for EC [15].

Materials and Methods

1

Comments 1:

[Remaining Issue: Although harmonization between “Damumed” and “Avicenna” systems is now described, it remains unclear how data mapping between categorical variables was validated statistically or manually. Please specify whether inter-database checks (e.g., frequency matching or random re-audit) were conducted to ensure consistency of coding standards.]

Response 1: Thank you for pointing this out.

[We thank the reviewer for this important methodological comment. In the revised manuscript, we have clarified how the harmonization of categorical variables between the Damumed and Avicenna systems was validated. Specifically, in the Data Sources section (Section 2.2) we now explain that a harmonized coding dictionary was created manually for variables that were originally coded differently in the two systems, based on their clinical meaning and documentation structure. After recoding, category distributions were reviewed to identify rare or potentially inconsistent values, and any ambiguities were resolved through manual re-checking of the original electronic medical records.

In addition, in the Data Completeness and Bias Control section (Section 2.4) we now state that inter-database consistency was further assessed by manually reviewing a random subset of records from both databases against the original electronic charts. These procedures did not reveal systematic discrepancies in category assignment between Damumed and Avicenna and ensured consistency of the harmonized dataset.]

This text was not in the article

Added a new text, more explanatory in section 2.2  of Data sources, in paragraph 4, in lines 200 to 208:

Data extraction and initial cleaning were performed by the research team using unified coding principles to ensure consistency between the two electronic systems. For categorical variables that were originally recorded differently in Damumed and Avicenna (e.g., education level, employment status, referral type, and selected comorbidities), a harmonized coding dictionary was created manually based on clinical meaning and documentation structure in both systems. After recoding, category distributions were reviewed to detect rare or potentially inconsistent values. When ambiguities were identified, the original electronic medical records were manually re-checked by the investigators to ensure correct category assignment.

Added a new text, more explanatory in section 2.4  of Data completeness and Bias Control, in paragraph 3, in lines 247 to 252:

To ensure inter-database consistency, the harmonized categorical variables were reviewed separately for each institution and study period at the stage of data prepara-tion. In addition, a random subset of records from both Damumed and Avicenna was manually compared with the original electronic charts to verify the correctness of re-coding and data transfer. This procedure did not reveal systematic discrepancies in category assignment between the two databases.

Results

1

Comments 1:

[1. Clarify in the Results whether all variables underwent completeness checks (particularly “smoking,” “BMI,” and “IUD use”) before inclusion in statistical analysis.]

Response 1: Thank you for pointing this out.

[We thank the reviewer for this important remark. In the revised manuscript, we have clarified in the Results section that all variables underwent completeness checks prior to statistical analysis, with particular attention to smoking status, BMI, and IUD use. We now explicitly state that records with missing values for specific variables were excluded only from the corresponding analyses and that all percentages were calculated based on the number of available observations for each parameter.]

This text was not in the article

Added a new text, more explanatory in the Results paragraph 1, in lines 311 to 316:

Prior to statistical analysis, all variables included in the Results section underwent completeness checks as part of the data preparation process. Particular attention was given to lifestyle- and history-related variables such as smoking status, body mass index (BMI), and intrauterine device (IUD) use. Records with missing values for specific variables were excluded only from the corresponding analyses, and all percentages were calculated based on the number of available observations for each parameter.

Comments 2:

[2.Although “asymptomatic” is now defined under Table 4, it would improve transparency to restate this operational definition briefly in the Materials and Methods section, not only as a footnote!!!]

Response 1: Thank you for pointing this out.

[We thank the reviewer for this helpful suggestion. In the revised manuscript, we have now explicitly restated the operational definition of “asymptomatic” in the Materials and Methods section, subsection 2.6 Study variables , in addition to the footnote in Table 4. We specify that “asymptomatic” refers to the absence of abnormal uterine bleeding, pelvic pain, or other gynecological complaints at the time of diagnosis, with endometrial hyperplasia detected incidentally during routine ultrasound examination or preventive gynecological check-up.]

This text was not in the article

Added a new text, more explanatory in the subsection 2.6 Study variables, paragraph 6, in lines 272 to 275:

In the present study, the term “asymptomatic” was operationally defined as the absence of patient-reported abnormal uterine bleeding, pelvic pain, or any other gyne-cological complaints at the time of diagnosis, with endometrial hyperplasia being de-tected incidentally during routine ultrasound examination or preventive gynecological check-up.

Discussion

1

Comments 1: [The new discussion is substantially more robust and logically organized. Subsections 4.1–4.4 reflect demographic, metabolic, and diagnostic transitions clearly. The new section cites several studies (e.g., Aro et al., Kim et al., Fader et al.) but does not include one comprehensive article summarizing HER2’s clinical significance across EC subtypes. Without this, the paragraph remains overly descriptive rather than integrative. Including this missing reference will strengthen scientific grounding and fulfill the original review request.

Metabolomic Profiling: The discussion still underrepresents metabolomic approaches as emerging diagnostic tools. The addition of 2–3 sentences summarizing the predictive potential of tissue and serum metabolomics in EH and early EC would make this section more complete and forward-looking.]

Response 1: Thank you for pointing this out.

[In response, we have further strengthened Section 4.5 (Current Research and Future Perspectives) to ensure a more integrative discussion of both HER2 and metabolomic approaches.

Regarding HER2, we have now incorporated a comprehensive contemporary clinical review summarizing the role of HER2 across aggressive endometrial cancer subtypes and its therapeutic implications (Papageorgiou, D.; Liouta, G.; Sapantzoglou, I.; Zachariou, E.; Pliakou, D.; Papakonstantinou, K.; Floros, T.; Pliakou, E. HER2-Positive Serous Endometrial Cancer Treatment: Current Clinical Practice and Future Directions. Medicina 2024, 60, 2012. https://doi.org/10.3390/medicina60122012). This addition provides an integrative clinical perspective that goes beyond individual molecular studies and fulfills the original request for a unifying reference.

Regarding metabolomic profiling, we have expanded the discussion by adding 2–3 sentences summarizing the predictive and diagnostic potential of tissue- and serum-based metabolomics in endometrial hyperplasia and early endometrial cancer. Specifically, we now cite a tissue-based untargeted metabolomic profiling study demonstrating discrimination between endometrial cancer, hyperplasia, and normal endometrium (Akkour, K.; Masood, A.; Al Mogren, M.; AlMalki, R.H.; Alfadda, A.A.; Joy, S.S.; Bassi, A.; Alhalal, H.; Arafah, M.; Othman, O.M.; et al. Tissue-Based Metabolomic Profiling of Endometrial Cancer and Hyperplasia. Metabolites 2025, 15, 458. https://doi.org/10.3390/metabo15070458), as well as a comprehensive review on metabolomic approaches for diagnosis and prognosis in endometrial cancer (Albertí-Valls, M.; Megino-Luque, C.; Macià, A.; Gatius, S.; Matias-Guiu, X.; Eritja, N. Metabolomic-Based Approaches for Endometrial Cancer Diagnosis and Prognosis: A Review. Cancers 2024, 16, 185. https://doi.org/10.3390/cancers16010185). These additions strengthen the forward-looking and translational dimension of the Discussion.]

This text was not in the article

We answered this question in the part where there were comments about adding missing links to the part of the Lead (described in detail above), and the discussion section was also discussed there.

Conclusions

1

Comments 1: [To further strengthen this section, add one final sentence underscoring that integration of biomarker and metabolomic data will be essential for future clinical translation, thereby linking back to the earlier missing molecular reference.

The revised manuscript demonstrates substantial progress and reflects genuine effort to address reviewer feedback. The structure, methodological transparency, and statistical reporting have markedly improved. However, publication cannot be recommended until required comments are fully addressed.]

Response 1: We thank the reviewer for this valuable final suggestion.

[In the revised manuscript, we have added a concluding sentence to the Conclusions section explicitly emphasizing that future integration of molecular biomarkers and metabolomic profiling into clinical registries and individualized patient assessment will be essential for successful clinical translation. This addition directly links the Conclusions to the molecular and metabolomic perspectives introduced earlier in the manuscript.]

This text was not in the article

Added in the 3rd paragraph of the conclusion section, the final sentences recommended by you:

In the future, the integration of molecular biomarkers and metabolomic profiling into population-based clinical registries and individualized patient assessment will be es-sential for the successful translation of epidemiological findings into precision preven-tion, early diagnosis, and personalized management of EH and EC. 

Round 4

Reviewer 1 Report

Comments and Suggestions for Authors

Thank you for the opportunity to read and review the revised version of your manuscript. The revised manuscript shows that the authors have carefully and systematically responded to the key points from previous peer review round. The molecular/biomarker context (HER2 and metabolomics), methods harmonization, variable completeness, and definition of “asymptomatic” are now clearly presented and supported with appropriate references. Scientifically, the manuscript is now much stronger and aligned with earlier requests.

Although a few minor issues still exist.

1.

Especially, in the main text (Section 2.8 Ethical approval) the authors correctly state that informed consent was not required due to anonymized retrospective data. In the “Statements” section near the end, they still have both: “separate informed consent was not required” and “Informed consent was obtained from all participants.”

2. 

There are a few minor grammatical errors throughout the manuscript (e.g. Line 580. Replace "have" with "has"). Please run a grammatical check.

Author Response

Response to Reviewer X Comments

Dear Reviewer,

On behalf of all co-authors — Imasheva Bayan Imashkyzy, Kamaliev Maksut Adilkhanovich, Lokshin Vyacheslav Notanovich, Kiseleva Marina Viktorovna, Laktionova Mariya Vladimirovna  — we would like to express our sincere gratitude for your valuable time, careful evaluation, and insightful comments on our manuscript entitled “Medical and Social Characteristics of Patients with Endometrial Hyperplasia in a Large City of Kazakhstan: A Retrospective Comparative Study.”

We deeply appreciate your constructive feedback and professional suggestions, which have greatly contributed to improving the clarity, structure, and scientific quality of our work. Each of your comments has been carefully considered and addressed in the revised version of the manuscript. Below, we provide a detailed point-by-point response highlighting all changes made in accordance with your recommendations.

With appreciation and respect,

Corresponding author — Bayan Imashkyzy Imasheva

Point-by-point response to Comments and Suggestions for Authors

Reviewer's comments

The authors' response

It was

Become

1

Comments 1: [1. Especially, in the main text (Section 2.8 Ethical approval) the authors correctly state that informed consent was not required due to anonymized retrospective data. In the “Statements” section near the end, they still have both: “separate informed consent was not required” and “Informed consent was obtained from all participants.”.]

Response 1: Thank you for pointing this out. [We fully agree that the statements regarding informed consent must be coherent throughout the manuscript. In accordance with the retrospective design of the study and the use of fully anonymized data, informed consent was not required. Therefore, we have revised the Informed Consent Statement to read:

“Informed Consent Statement: Informed consent was not required due to the retrospective design of the study and the use of fully anonymized data.”

This revision eliminates the contradiction between Section 2.8 (Ethical approval) and the Statements section.]

In terms of the Statements section, these changes are reflected in lines 586 to 587

The old text of the description:

Statement of informed consent:

Informed consent was obtained from all participants in the study.

New corrected text:

In terms of the Statements section, these changes are reflected in lines 586 to 587:

Statement of informed consent: Informed consent was not required due to the retrospective design of the study and the use of fully anonymized data.

2

Comments 2: [2.There are a few minor grammatical errors throughout the manuscript (e.g. Line 580. Replace "have" with "has"). Please run a grammatical check.

Response 1: Thank you for pointing this out.

[We appreciate this remark. We have carefully re-read the manuscript and corrected minor grammatical issues throughout the text. In particular, we changed “Finding: This study have not received any external funding” to “Funding: This study has not received any external funding.”, and corrected formulations such as “can reliable distinguish” to “can reliably distinguish” and “In large implementation study” to “In a large implementation study”. “CCH No 7” to “City Clinical Hospital No. 7”. We have also refined several additional sentences for clarity and grammatical accuracy.]

Response to Reviewer X Comments

Dear Reviewer,

On behalf of all co-authors — Imasheva Bayan Imashkyzy, Kamaliev Maksut Adilkhanovich, Lokshin Vyacheslav Notanovich, Kiseleva Marina Viktorovna, Laktionova Mariya Vladimirovna  — we would like to express our sincere gratitude for your valuable time, careful evaluation, and insightful comments on our manuscript entitled “Medical and Social Characteristics of Patients with Endometrial Hyperplasia in a Large City of Kazakhstan: A Retrospective Comparative Study.”

We deeply appreciate your constructive feedback and professional suggestions, which have greatly contributed to improving the clarity, structure, and scientific quality of our work. Each of your comments has been carefully considered and addressed in the revised version of the manuscript. Below, we provide a detailed point-by-point response highlighting all changes made in accordance with your recommendations.

With appreciation and respect,

Corresponding author — Bayan Imashkyzy Imasheva

Point-by-point response to Comments and Suggestions for Authors

Reviewer's comments

The authors' response

It was

Become

1

Comments 1: [1. Especially, in the main text (Section 2.8 Ethical approval) the authors correctly state that informed consent was not required due to anonymized retrospective data. In the “Statements” section near the end, they still have both: “separate informed consent was not required” and “Informed consent was obtained from all participants.”.]

Response 1: Thank you for pointing this out. [We fully agree that the statements regarding informed consent must be coherent throughout the manuscript. In accordance with the retrospective design of the study and the use of fully anonymized data, informed consent was not required. Therefore, we have revised the Informed Consent Statement to read:

“Informed Consent Statement: Informed consent was not required due to the retrospective design of the study and the use of fully anonymized data.”

This revision eliminates the contradiction between Section 2.8 (Ethical approval) and the Statements section.]

In terms of the Statements section, these changes are reflected in lines 586 to 587

The old text of the description:

Statement of informed consent:

Informed consent was obtained from all participants in the study.

New corrected text:

In terms of the Statements section, these changes are reflected in lines 586 to 587:

Statement of informed consent: Informed consent was not required due to the retrospective design of the study and the use of fully anonymized data.

2

Comments 2: [2.There are a few minor grammatical errors throughout the manuscript (e.g. Line 580. Replace "have" with "has"). Please run a grammatical check.

Response 1: Thank you for pointing this out.

[We appreciate this remark. We have carefully re-read the manuscript and corrected minor grammatical issues throughout the text. In particular, we changed “Finding: This study have not received any external funding” to “Funding: This study has not received any external funding.”, and corrected formulations such as “can reliable distinguish” to “can reliably distinguish” and “In large implementation study” to “In a large implementation study”. “CCH No 7” to “City Clinical Hospital No. 7”. We have also refined several additional sentences for clarity and grammatical accuracy.]